# Analogue modelling of basin inversion: implications for the Araripe Basin (Brazil)

**Pâmela C. Richetti**[1,2]**, Frank Zwaan**[2,3,4]**, Guido Schreurs**[2]**, Renata S. Schmitt**[1,2,5]**, and Timothy C. Schmid**[2]

[1]Programa de Pós-graduação em Geologia – PPGL, Universidade Federal do Rio de Janeiro, Rio de Janeiro, Brazil
[2]Institute of Geological Sciences, University of Bern, Bern, Switzerland
[3]Helmholtz Centre Potsdam, GFZ German Research Centre for Geosciences, Potsdam, Germany
[4]Department of Geosciences, University of Fribourg, Fribourg, Switzerland
[5]Departamento de Geologia – IGEO, Universidade Federal do Rio de Janeiro, Rio de Janeiro, Brazil

**Correspondence:** Pâmela C. Richetti (pamelarichetti@geologia.ufrj.br)

**Abstract.** Basin inversion is a process that takes place when a sedimentary basin is subjected to compressional stresses resulting in the reactivation of pre-existing faults and/or the localization of deformation along new reverse faults. The Araripe Basin (NE Brazil) is an example of a Cretaceous intracontinental aborted rift, with its sedimentary infill found at ca. 1000 m altitude, 500 m above the host basement. Post-rift basin inversion has been proposed by previous authors as the cause of this topographic high, but how inversion affected this basin remains a matter of debate, with the following two end-member scenarios: reactivation of pre-existing normal faults leading to local uplift or regional uplift and differential erosion. Neither end-member fully explains the observations from seismic and field data. In this study, we, therefore, conducted analogue models to explore how basin inversion in the Araripe Basin could have taken place. We present two series of crustal-scale brittle–viscous experiments: (i) extension followed by compression without sedimentation, with a variation in divergence and convergence directions (orthogonal or 45° oblique); and (ii) extension with syn-rift sedimentation followed by compression, with the same variation in rifting and inversion directions. We found that orthogonal rifting without sedimentation forms throughgoing graben boundary faults, whereas oblique rifting initially creates en échelon faults that eventually link up, creating large graben boundary faults. Rift basins with syn-rift sedimentation evolved in a similar fashion; however, sedimentary loading resulted in increased subsidence. During both oblique and orthogonal inversion, most shortening was accommodated along new low-angle reverse faults. Significant intra-graben fault reactivation occurred in all models without syn-rift sedimentation. By contrast, orthogonal inversion of models with syn-rift sedimentation did not reactivate rift faults, whereas only a minor reactivation of rift faults took place during oblique inversion since the sediments strengthened the otherwise weakened basin, thus acting as a buffer during convergence. Based on our modelling results, we propose an alternative scenario for the evolution of the Araripe Basin, involving oblique inversion and the development of low-angle reverse faults, which better fits observations from seismic lines and field data from the region.

## 1 Introduction

The inversion of sedimentary basins as a result of compressional tectonics is a widely discussed topic due to its importance for the development of mineral and hydrocarbon deposits (Sibson and Scott, 1998; Turner and Williams, 2004). In particular, inverted intraplate rift basins that are currently exposed above sea level can play an important role in the understanding of their offshore equivalents, since they provide access to outcrops that otherwise can only be analysed via indirect geophysical methods (e.g. Stanton et al., 2014; Rebelo et al., 2021).

In this context, the Araripe Basin in NE Brazil is an excellent example of an exposed inverted intraplate rift basin (Fig. 1). This Early Cretaceous rift basin is part of the aborted

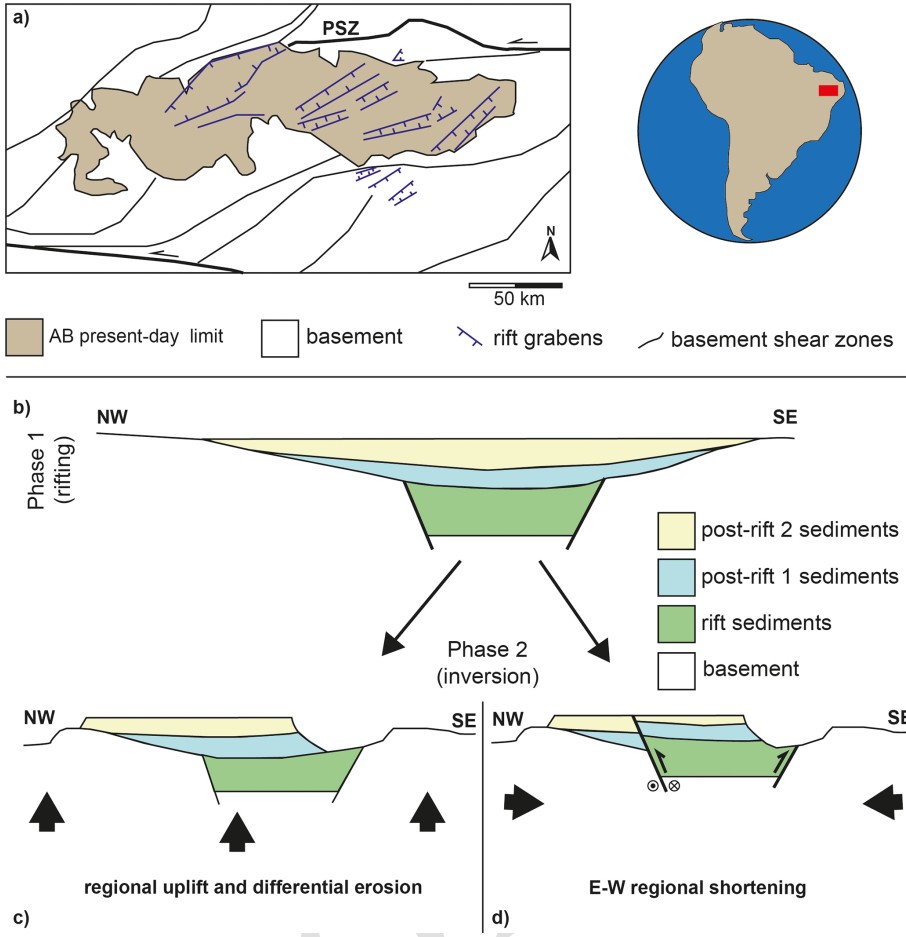

**Figure 1. (a)** Structural geology of the study area and present-day Araripe Basin (AB), depicting NE–SW-striking rift-related structures (in blue) and Precambrian basement shear zones (in black), after Camacho and Souza (2017). Note that most faults (in blue) in the Araripe Basin are covered by post-rift sediments and are interpreted from reflection seismic sections (Ponte and Ponte-Filho, 1996). PSZ is the Patos shear zone. **(b)** Schematic NW–SE section representing rift and post-rift formations in the Araripe Basin prior to inversion. **(c)** Schematic representation of the Araripe Basin inversion model based on regional uplift followed by differential erosion proposed by Peulvast and Bétard (2015). **(d)** Schematic representation of the Araripe Basin inversion model as a result of regional oblique convergence proposed by Marques et al. (2014). TS1

northeast Brazilian rift system (NBRS; de Matos, 1992), which is located at the intersection of the equatorial and central segments of the South Atlantic Ocean (Moulin et al., 2010). This rift system formed within the well-developed network of NE–SW- and E–W-striking Precambrian ductile shear zones in the basement of the Borborema Province (Fig. 1a; Vauchez et al., 1995; Brito Neves et al., 2000; Ganade de Araujo et al., 2014). The rift structures within the E–W-oriented Araripe Basin mainly strike NE–SW (Fig. 1a), indicating brittle reactivation of the basement shear zones during rifting (de Matos, 1992). However, the exact kinematics of rifting during Araripe Basin formation remains a matter of debate, with some authors proposing orthogonal kinematics, whereas others invoke transtension (e.g. Rosa et al., 2023).

After rifting and subsequent thermal subsidence (Assine, 2007), the basin registered a phase of inversion (Marques et al., 2014; Peulvast and Bétard, 2015), and its sedimentary infill is presently situated, at its highest point, at 1000 m above sea level and ca. 500 m above the surrounding basement. Similarly, the Borborema Province generally contains high topographies and evidence of recent uplift (Lamarque and Julià, 2019; Neto et al., 2019), and other basins in the NBRS also present evidence of tectonic inversion (Gurgel et al., 2013; Nogueira et al., 2015; Vasconcelos et al., 2021; Bezerra et al., 2020; Ramos et al., 2022). In the Araripe Basin, Marques et al. (2014) proposed that inversion resulted from far-field ENE–WSW-directed horizontal maximum compressive stress (Fig. 1d). They concluded that this deformation is consistent with the formation of new oceanic crust in the South Atlantic to the east and the development of

the Andes to the west, resulting in the overall compression of the South American Plate (Coblentz and Richardson, 1996; Marques et al., 2013).

According to Marques et al. (2014), this compression caused a large-scale inversion of the initial high-angle normal faults of the Araripe Basin (Fig. 1d) through the oblique convergence and injection of soft material into these faults. By contrast, Peulvast and Bétard (2015) proposed that the present-day topographic elevation of the basin is due to regional uplift of the Borborema Province and the action of differential erosion (Fig. 1c). The Peulvast and Bétard (2015) scenario fits with the general absence of large-scale inversion of normal faults, as seen on seismic sections from the Araripe Basin (Ponte and Ponte-Filho, 1996; Rosa et al., 2023). However, on closer inspection, these seismic sections do in fact show a limited degree of normal fault inversion (Ponte and Ponte-Filho, 1996; Cardoso, 2010; Rosa et al., 2023), and localized reverse faulting linked to basin inversion is observed in nearby basins of the same age as well (e.g. the Rio do Peixe Basin; Vasconcelos et al., 2021). As such, the exact mechanism causing inversion and to what degree rift structures were reactivated in the Araripe Basin remains unclear, thus requiring further research with additional approaches. One of these approaches is analogue tectonic modelling, which has shown to be a useful tool for understanding the evolution of inverted basins and the mechanisms involved in various settings (Brun and Nalpas, 1996; Nalpas et al., 1995; Panien et al., 2005; del Ventisette et al., 2005, 2006; Marques and Nogueira, 2008; Pinto et al., 2010; di Domenica et al., 2014; Jara et al., 2018; Zwaan et al., 2022b).

In this paper, we therefore present the results of new crustal-scale analogue tectonic modelling experiments completed with a novel set-up, which were aimed at evaluating whether tectonic compression could have caused the inversion observed in the Araripe Basin. In our models, we tested the general influence of orthogonal ($\alpha = 0$) or oblique ($\alpha = 45°$) divergence, followed by either orthogonal or oblique convergence, and syn-rift sedimentation on initial basin development and on subsequent inversion structures. We subsequently compare our first-order model results with data from nature and propose an updated scenario for inversion of the Araripe Basin involving oblique inversion and the development of low-angle reverse faults outside the basin.

## 2 Methods

### 2.1 Model set-up

For this study of the crustal-scale basin inversion processes, we used an experimental set-up involving two long mobile sidewalls, two rubber end walls (fixed between the mobile sidewalls and closing the short model ends), and a base consisting of a mobile and a fixed base plate (Fig. 2a). We positioned a 5 cm thick block consisting of intercalated foam and plexiglass bars (each 1 and 0.5 cm wide, respectively) above the base plates and between the long sidewalls (Fig. 2a, b). This foam and plexiglass block, initially 36.5 cm wide, was compressed prior to adding the model materials in order to reach the initial width of 30 cm (Fig. 2a, b). Divergence of the mobile long sidewalls, achieved by high-precision computer-controlled motors, simulated an initial rifting phase inducing uniform orthogonal divergence into the overlying brittle and viscous model materials that represent the brittle upper crust and ductile lower crust, respectively (see also Sect. 2.2). For orthogonal convergence during the subsequent inversion phase, the sidewalls were simply moved together again. During oblique divergence and oblique convergence, which we applied to account for possible different deformation kinematics during basin formation and inversion, such as proposed by, for example, Marques et al. (2014) and Rosa et al. (2023), the additional lateral motion of one mobile base plate was introduced (Fig. 2c). In order to localize deformation in our models and to create a graben during the initial rifting phase, we inserted a linear seed, which was made from the same viscous material as used for the simulated lower-crustal layer at the base of the brittle sand cover representing the upper crust (e.g. Le Calvez and Vendeville, 2002; Molnar et al., 2019, 2020; Zwaan and Schreurs, 2017). This seed was a semi-cylindrical ridge with a ca. 1 cm diameter and was placed in the same position in each model (i.e. along the central axis of the model; Fig. 2a, b).

Our general model set-up has been regularly used for the simulation of orthogonal and oblique rifting, as well as transpressional tectonics (Schreurs and Colletta, 1998, 2002; Zwaan and Schreurs, 2017; Zwaan et al., 2016, 2018a, 2020; Schmid et al., 2022). However, so far only Guillaume et al. (2022) have applied a similar foam-based set-up for basin inversion modelling, with the key difference being that the convergence direction in their models was perpendicular to the divergence direction. Our model set-up design is also fundamentally different from previous basin inversion model set-ups involving base plates and/or sidewalls for simulating orthogonal and oblique basin inversion, which tend to strongly localize model deformation along the base plate edges or at the sidewalls, respectively (e.g. Brun and Nalpas, 1996; Nalpas et al., 1995; see also Zwaan et al., 2022b, for an extensive discussion on analogue basin inversion model set-ups). We also note that our current model set-up is well suited to reproducing the large-scale structures that may develop in inverted rift basins such as the Araripe Basin but may not capture all peculiarities of the specific natural example. As such, our comparison with the Araripe Basin must remain on a first-order scale. Even so, we believe our model results suffice to address the scenarios for proposed inversion of the Araripe Basin, since these scenarios also concern the large-scale structural evolution of the basin.

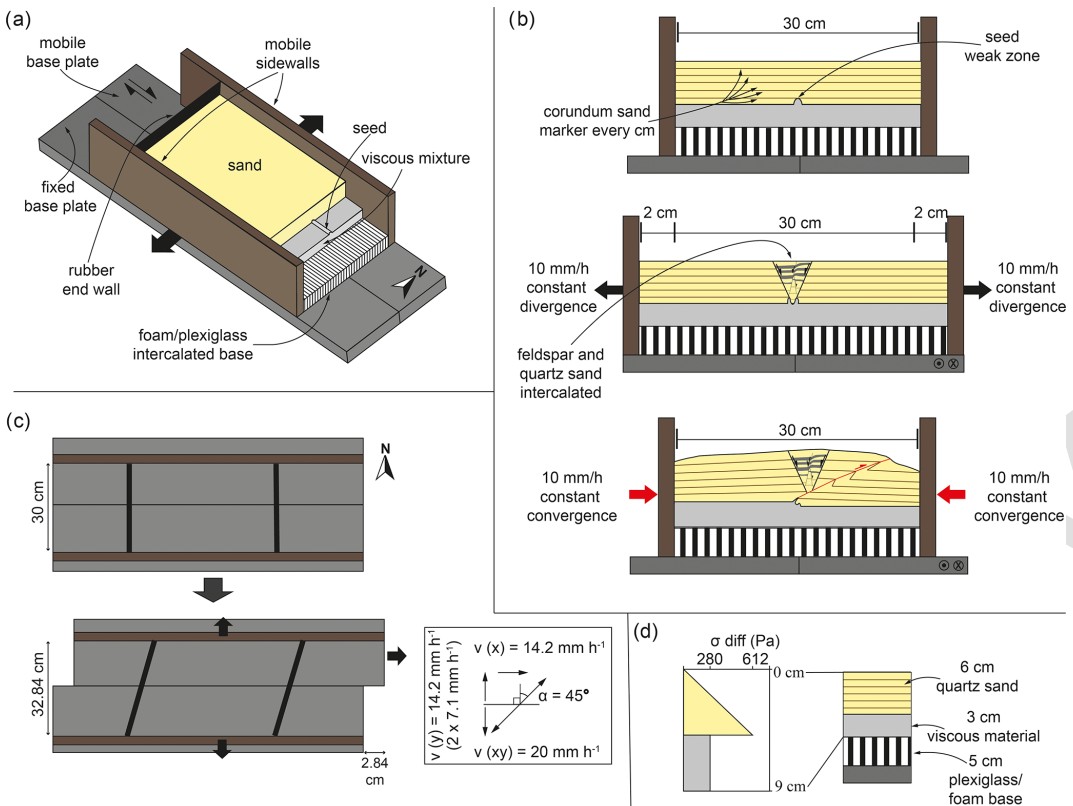

**Figure 2.** Experimental set-up adopted for this study. **(a)** 3D cut-out view showing the brittle–viscous layers on top of the plexiglass/foam base of the experiment (north arrow added for reference in the models). **(b)** Schematic example of a sedimentation model run in 2D. **(c)** Top view example of movement direction of the experimental apparatus used in this study (oblique divergence example, with the definition of divergence and convergence obliquity given as angle $\alpha$. Note that angle $\alpha$ is positive for dextral oblique divergence and negative for sinistral oblique convergence. Conversely, angle $\alpha$ is negative for sinistral oblique divergence and positive for dextral oblique convergence). **(d)** Schematic strength profile indicating the crustal setting represented in our models.

## 2.2 Materials

We utilized brittle and viscous analogue materials (material properties are summarized in Table 1) to reproduce the brittle and ductile parts of the upper and lower crust in our experiments.

A 3 cm thick viscous layer served to replicate a 10 km thick lower crust. This material consists of a near-Newtonian ($\eta$ = ca. $1.5 \times 10^5$ Pa s; $n = 1.05$–$1.10$; Zwaan et al., 2018c) mixture of SGM-36 polydimethylsiloxane (PDMS) and corundum sand ($\rho_{\text{bulk material}} = 3950$ kg m$^{-3}$; Carlo AG, 2023). We mixed the components according to a 0.965 : 1.00 weight ratio, resulting in a viscous mixture with a density of ca. 1600 kg m$^{-3}$.

We applied a 6 cm thick layer of fine quartz sand ($\varnothing$ = 60–250 µm and $\phi = 31.4$–$36.1°$; Zwaan et al., 2018a) sieved on top of the viscous layer, representing a 20 km brittle upper crust. During model preparation, the sand was flattened at 1 cm intervals with a scraper to avoid lateral variations in the sand layer thickness. We furthermore sieved the sand from ca. 30 cm height to ensure a constant brittle layer density

of ca. 1560 kg m$^{-3}$ (e.g. Klinkmüller et al., 2016; Schmid et al., 2020).

We adopted layers of feldspar sand (grain size range = 100–250 µm and $\phi = 29.9$–$35°$; Zwaan et al., 2022c) intercalated with layers of the same quartz sand used for the crustal layer to simulate sedimentary infill, where the intercalation served to provide a visual record of syn-rift units on cross sections (Fig. 2b). The simulated sedimentary infill was manually applied, using a paper cone with an opening of 3 mm at the tip. The flux of sand representing the sediments was controlled by pressing the opening of the cone, and we filled the graben up to the general model surface.

Furthermore, we added thin < 1 mm thick marker intervals of fine corundum sand (grain size range = 88–125 µm) to the quartz sand layer representing the upper crust, which allowed for the tracing of deformation in section view (Fig. 2b). These thin intervals were sieved in during the scraping intervals (every centimetre) and are not considered to have an impact on model evolution.

**Table 1.** Material properties.

| Granular materials | Quartz sand[a] | Corundum sand[b] | Feldspar sand[h] |
|---|---|---|---|
| Grain size range ($\varnothing$) | 60–250 µm | 88–125 µm | 100–250 µm |
| Bulk material density ($\rho_{\text{bulk material}}$)[c] | 2650 kg m$^{-3}$ | 3950 kg m$^{-3}$ | ca. 2700 kg m$^{-3}$ |
| Sieved density ($\rho_{\text{sieved}}$) | 1560 kg m$^{-3}$ | 1890 kg m$^{-3}$ | ca. 1300 kg m$^{-3}$ |
| Angle of internal peak friction ($\phi_{\text{peak}}$) | 36.1° | 37° | 35° |
| Coefficient of internal peak friction ($\mu_{\text{peak}}$)[d] | 0.73 | 0.75 | 0.70 |
| Angle of dynamic–stable friction ($\phi_{\text{dyn}}$) | 31.4° | 32.0° | 29.9° |
| Coefficient of dynamic–stable friction ($\mu_{\text{dyn}}$)[d] | 0.66 TS2 | 0.62 | 0.58 |
| Angle of reactivation friction ($\phi_{\text{react}}$) | 33.5° | – | 32.0° |
| Coefficient of reactivation friction ($\mu_{\text{react}}$) | 0.66 | – | 0.62 |
| Cohesion ($C$) | 9 ± 98 Pa | 39 ± 10 Pa | 51 Pa |
| Viscous material | Pure polydimethylsiloxane (PDMS)[a, e] | PDMS/corundum sand mixture[a] | |
| Weight ratio PDMS : corundum sand | – | 0.965 kg : 1.00 kg | |
| Density ($\rho$) | 965 kg m$^{-3}$ | ca. 1600 kg m$^{-3}$ | |
| Viscosity ($\eta$) | ca. $2.8 \times 10^4$ Pa s | ca. $1.5 \times 10^5$ Pa s[f] | |
| Type[f] | Newtonian ($n =$ ca. 1)[g] | Near-Newtonian ($n = 1.05$–$1.10$)[g] | |

[a] Quartz sand, PDMS, and viscous mixture characteristics are from Zwaan et al. (2016, 2018a, b). [b] Corundum sand characteristics are from Panien et al. (2006). [c] Bulk material densities are from Carlo AG (2023). [d] $\mu = \tan(\phi)$. [e] Pure PDMS rheology details are from Rudolf et al. (2016). [f] The viscosity value holds for model strain rates $< 10^{-4}$ s$^{-1}$. [g] The power law exponent $n$ (dimensionless) represents sensitivity to strain rate. [h] Feldspar sand characteristics are from Zwaan et al. (2022c).

## 2.3 Model parameters

For this study, we completed two main series of four experiments each and an initial series of reference experiments (Table 2). Series A contains our reference experiments that simulated the initial (orthogonal) rifting phase only, with and without syn-rift sedimentation. Series B explores the effects of basin inversion without syn-rift sedimentation. Series C tests the effects of syn-rift sedimentation during basin inversion. The initial rifting phase of our series B and C basin inversion models involved either orthogonal or 45° oblique divergence (where obliquity is defined by angle $\alpha$; i.e. the angle between the normal to the rift axis and the divergence direction; Fig. 2c). The subsequent phase of shortening involved either orthogonal or $(-)$45° oblique convergence (see details in Table 2). The experiments ran for 2 h, with 40 mm of divergence (at 20 mm h$^{-1}$) and another 2 h with 40 mm of convergence, except for models B3 and C3, since the initial oblique opening did not generate sufficient space for the subsequent 40 mm of orthogonal convergence. Therefore, total convergence in models B3 and C3 amounted to 28 mm over an interval of 85 min instead, which was, however, a sufficient level of convergence to establish well-developed inversion features.

We implemented syn-rift sedimentation in five of our experiments (in model A2 and in models C1–4), by halting the machine every 15 min (eight sedimentary intervals in total for 2 h of rifting) and filling the accommodation space manually with feldspar and quartz sand in alternating intervals (Fig. 2b; see also Sect. 2.2). The two experiments with oblique rifting had only seven sedimentation intervals because after the first 15 min, insufficient accommodation space was generated, requiring us to start the first sand filling after 30 min instead. In each model, the final sedimentation interval after the end of rifting generated a nearly flat model topography prior to inversion (Fig. 1b).

## 2.4 Scaling

Model scaling is important to guarantee that experiments completed in the laboratory are representative of their counterparts in nature. For the brittle materials, the main parameter is the angle of internal friction (35–37°), which is similar to the internal friction angle values found for upper-crustal rocks (31–38°; Byerlee, 1978; Table 3). In order to scale the viscous material, we must consider its strain-rate-dependent rheology. The stress ratio between model and nature ($\sigma^*$, with a convention of $\sigma^* = \sigma_{\text{model}}/\sigma_{\text{nature}}$) is calculated as follows: $\sigma^* = \rho^* \cdot h^* \cdot g^*$, where $\rho^*$, $h^*$, and $g^*$ represent density, length, and gravity ratios, respectively (Hubbert, 1937; Ramberg, 1981). Combined with the viscosity ratio ($\eta^*$), the stress ratio yields the strain rate ratio $\acute{\varepsilon}^*$ (Weijermars and Schmeling, 1986) as follows: $\acute{\varepsilon}^* = \sigma^*/\eta^*$. Subsequently, the velocity and time ratios ($v^*$ and $t^*$) are derived from the strain rate ratio: $\acute{\varepsilon}^* = v^*/h^* = 1/t^*$. We adopt a relatively high lower-crustal viscosity of ca. $5 \times 10^{21}$, representing a typical early magma-poor rift system (e.g. Buck, 1991). Thus, 1 h in our model represents ca. 1.3 Myr in nature, and 20 mm h$^{-1}$ of divergence/convergence in the model

**Table 2.** Parameters of analogue models performed in this study.

| Model series | Model name | Direction and velocity of divergence/convergence | | | | Sedimentation | Sections made |
| --- | --- | --- | --- | --- | --- | --- | --- |
| | | Phase 1 (40 mm of divergence) | | Phase 2 (40 mm of convergence) | | | |
| | | Direction (angle $\alpha$) | Velocity ($v$) (mm h$^{-1}$) | Direction (angle $\alpha$) | Velocity ($v$) (mm h$^{-1}$) | | |
| Series A | A1 | 0° | 20 | – | – | No | Yes |
| Reference rifting models | A2 | 0° | 20 | – | – | Yes | Yes |
| Series B | B1 | 0° | 20 | 0° | 20 | No | Yes[a] |
| Rifting and inversion | B2[b] | 0° | 20 | −45° | 20 | No | No |
| without | B3[c] | 45° | 20 | 0° | 20 | No | No |
| sedimentation | B4 | 45° | 20 | 45° | 20 | No | No |
| Series C | C1 | 0° | 20 | 0° | 20 | Yes | Yes |
| Rifting and | C2[b] | 0° | 20 | −45° | 20 | Yes | Yes |
| inversion with | C3[c] | 45° | 20 | 0° | 20 | Yes | Yes |
| sedimentation | C4 | 45° | 20 | 45° | 20 | Yes | Yes |

[a] Sections are not used in this paper but are presented in the Supplement (Richetti et al., 2023). [b] Models with initial orthogonal divergence underwent dextral inversion ($\alpha = -45°$) due to the technical limitations of our model apparatus. However, one can simply mirror the result to obtain the sinistral inversion equivalent ($\alpha = 45°$). [c] Models with reduced inversion duration due to the oblique divergence with a reduced orthogonal divergence component.

embodies a realistic deformation velocity of ca. 5 mm yr$^{-1}$ in nature. The scaling parameters are presented in Table 3.

The dynamic similarity of the model and natural example can also be examined. First, the dynamic similarity between the model brittle layer and its upper-crustal equivalent can be determined through the ratio $R_s$ between the gravitational stress and the cohesive strength or cohesion $C$ (Ramberg, 1981; Mulugeta, 1988) as follows: $R_s = (\rho \cdot g \cdot h)/C$. The 9 Pa cohesion in the sand and a natural cohesion of 5 MPa for upper-crustal rocks, gives us a $R_s$ value of 102 and 110 for model and nature, respectively. Second, the dynamic similarity between our viscous material and lower-crust equivalent is derived from the Ramberg number $Rm$, which defines the ratio of gravitational stress to viscous strength (Weijermars and Schmeling, 1986), as follows: $Rm = (\rho \cdot g \cdot h^2)/(\eta \cdot v)$, and both have a value of 68. We consider our models to be properly scaled for simulating crustal-scale inversion processes, since their $R_s$ and $R_m$ values are similar to their natural equivalent.

## 2.5 Model monitoring and analysis

The experiments were primarily monitored through time-lapse photographs of the model surface, with pictures taken every minute for the duration of the model run. One central camera (Nikon D810; 36 MPx) provided map view pictures, while two obliquely oriented cameras (Nikon D810; 36 MPx) were positioned on both sides of the central one to provide stereoscopic imagery. This central camera was controlled using Nikon Camera Control Pro software, and cameras for stereoscopic imagery were remotely triggered by passing on

the signal from the central camera via an ESPER TriggerBox (Schmid et al., 2022).

To facilitate the first-order surface deformation analysis, we sieved a thin grid (4 × 4 cm) of corundum sand on the model surface. We furthermore sprinkled the model surface with coffee powder to provide markers for later digital image correlation (DIC) analysis. For the models involving syn-rift sedimentation, a fine layer (< 1 mm) of quartz sand was sieved on the top of the experiment at the end of rifting phase to create a blank surface with a new grid and new coffee markers, allowing for optimal tracing of the deformation during the inversion phase. Note that we defined a north reference in the models in order to facilitate the description of our model results (Fig. 2)

To quantify and visualize the surface deformation evolution of the experiments, we applied a detailed analysis of the time-lapse photographs through DIC techniques (e.g. Adam et al., 2005; Boutelier et al., 2019; Marshak et al., 2019; Zwaan et al., 2021; Schmid et al., 2022). The DIC analysis was performed by comparing the top-view images of subsequent time steps using LaVision's DaVis software (version 10.2). We used a calibration plate with a cross pattern of known dimensions as a reference to unwarp and rectify images and to scale calculated displacements. Incremental maximum and minimum normal strains are defined as the magnitude of the largest (i.e. stretching) and smallest (i.e. shortening) axes of the strain ellipse and are independent of reference frame (e.g. Broerse et al., 2021). These strains are therefore suitable markers to trace and quantify active extension and shortening (i.e. faults) in our experiments, respectively.

**Table 3.** Scaling parameters.

| | | Model | Nature |
|---|---|---|---|
| General parameters | Gravitational acceleration ($g$) | $9.81 \, \mathrm{m\,s^{-2}}$ | $9.81 \, \mathrm{m\,s^{-2}}$ |
| | Divergence velocity ($v$) | $5.6 \times 10^{-6} \, \mathrm{m\,s^{-1}}$ | $1.7 \times 10^{-10} \, \mathrm{m\,s^{-1}}$ |
| Brittle layer | Material | Quartz sand | Upper crust |
| | Peak internal friction angle | 35–37° | 31–38° |
| | Thickness ($h$) | $6 \times 10^{-2} \, \mathrm{m}$ | $2 \times 10^4 \, \mathrm{m}$ |
| | Density ($\rho$) | $1560 \, \mathrm{kg\,m^{-3}}$ | $2800 \, \mathrm{kg\,m^{-3}}$ |
| | Cohesion ($C$) | $9 \, \mathrm{Pa}$ | $5 \times 10^6 \, \mathrm{Pa}$ |
| Viscous/ductile layer | Material | PDMS–corundum sand mixture | Lower crust |
| | Thickness ($h$) | $3 \times 10^{-2} \, \mathrm{m}$ | $1 \times 10^4 \, \mathrm{m}$ |
| | Density ($\rho$) | $1600 \, \mathrm{kg\,m^{-3}}$ | $2900 \, \mathrm{kg\,m^{-3}}$ |
| | Viscosity ($\eta$) | $1.5 \times 10^5 \, \mathrm{Pa\,s}$ | $1 \times 10^{21} \, \mathrm{Pa\,s}$ |
| Dynamic scaling values | Brittle stress ratio ($R_\mathrm{s}$) | 102 | 110 |
| | Ramberg number ($R_\mathrm{m}$) | 68 | 68 |

To reconstruct the model topography in detail, we used the pairs of high-resolution oblique photographs taken at 30 min time steps. Agisoft PhotoScan photogrammetry software served to merge these pairs of synchronous photographs through the use of markers with known coordinates in the experiment for georeferencing, allowing us to create detailed digital elevation models (DEMs). These DEMs, shown in map view, and the extracted topography profiles over time are combined with the DIC results for a complete interpretation of model surface evolution (e.g. Maestrelli et al., 2020; Zwaan et al., 2022a).

Finally, cross-sections were made to reveal the internal structures of the models at the end of the model run (i.e. at the end of the rifting phase for series A models and after inversion for series B and C). In order to produce these sections, we added water with soap to the edges of the model until the sand was saturated and stable, and for every model, we cut six sections orthogonal to the model axis, each 10 cm apart. Pictures were taken for an analysis of the internal structures and for the quantification of subsidence. The sections of the reference models (series A) provide insights into the graben structures prior to inversion.

## 3 Results

The results of our model analysis are presented in summary figures for each experiment (Figs. 3–8). We show the incremental maximum and minimum normal strain from the DIC analysis results in map view for the early stage (at $t = 30$ min) and end stage of each phase (at $t = 120$ (or 85) min), topography maps at the end of each deformation phase, and topographic profiles over 30 min increments. Moreover, model sections are presented for series A and C.

### 3.1 Series A – reference models

The series A models provided a reference for the series B and C analysis. These models had a constant orthogonal divergence direction ($\alpha = 0°$) and a divergence velocity of $20 \, \mathrm{mm\,h^{-1}}$ (Fig. 3). In model A1, no sedimentation was applied during rifting, whereas in model A2, eight phases of syn-rift sedimentation were applied at 15 min intervals.

#### 3.1.1 Orthogonal rift without syn-rift sedimentation – model A1

The deformation in model A1 is localized in the first 30 min (Fig. 3a), with two graben boundary faults rooting in the viscous seed (Fig. 3i) accommodating the extension in one E–W-striking graben. Towards the end of the rifting phase ($t = 120$ min; Fig. 3b), a second-generation intra-graben fault developed between the two conjugate graben boundary faults. The strain analysis indicates higher strain values in the southern graben border fault and within the second-generation intra-graben fault (Fig. 3b). However, the northern graben border fault also remained active until the end of the experiment (Fig. 3b). Sections show the drag folds associated with the northern and southern graben boundary faults (Fig. 3i). The final topography profiles (Fig. 3d; $t = 60, 90,$ and 120 min) show a V-shaped depression on the southern side of the graben floor. This topographic feature can be related to the drag fold of the southern graben block seen in the section view CE1 (Fig. 3i, panel II), indicating that the drag fold was initiated after the first hour of the experiment and continued evolving until the end of the rifting phase. In Fig. 3i (panel II), we measured graben width between the two master faults bounding the grabens, which yielded a width of 56.2 mm. To measure the total vertical fault offset, we used

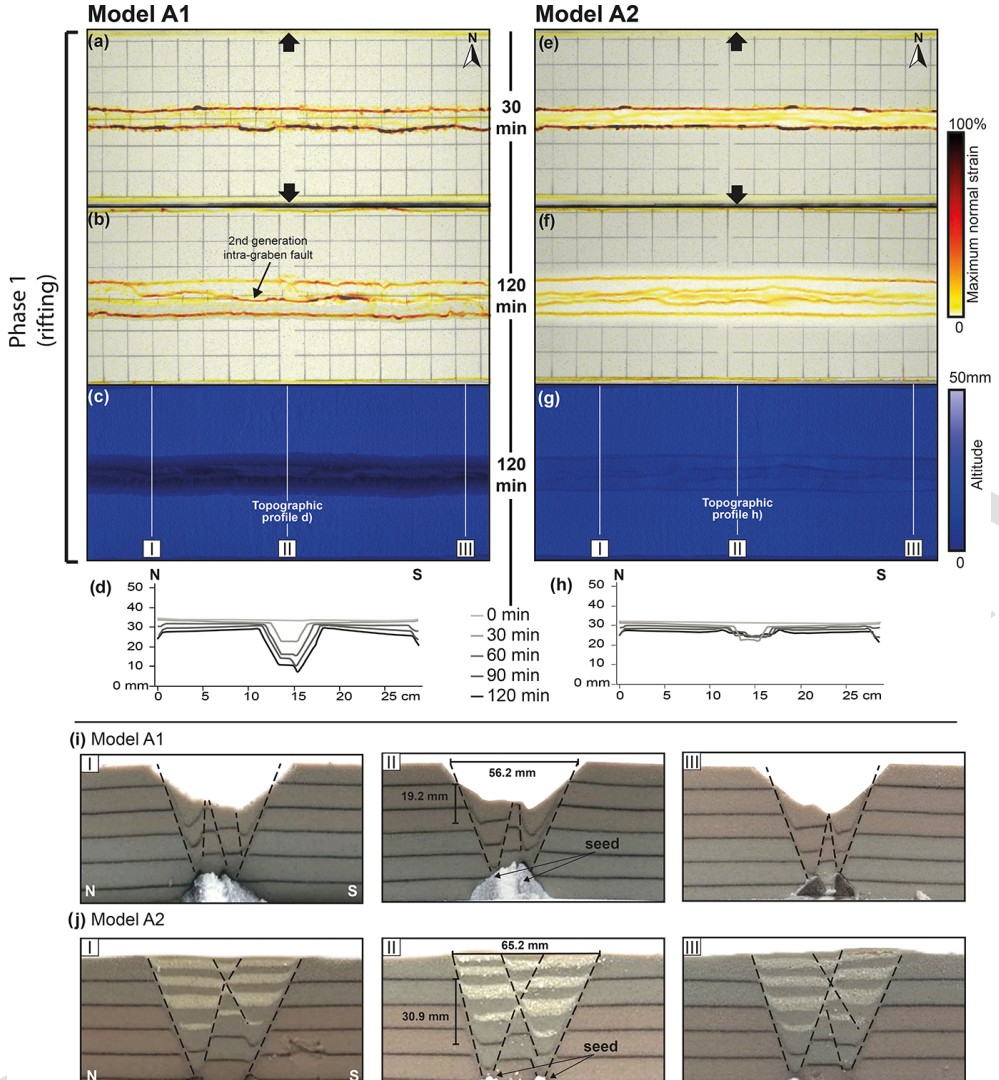

**Figure 3.** Evolution of deformation during orthogonal rifting for models A1 (no sedimentation) and A2 (with sedimentation). **(a, b, e, f)** Top-view incremental maximum normal strain results for early- and late-stage rifting, respectively, projected on greyscale top view imagery of the model surface. **(c, g)** Top views of digital elevation models at the end of rifting. **(d, h)** Topographic profiles for every 30 min of rifting. The vertical exaggeration is 4. Note that in model A2, topography is shown prior to syn-rift sedimentation for each time interval. **(i, j)** Sections for models A1 and A2, respectively. **(c, g)** The section locations. Graben geometry measurements are provided in the middle sections (corresponding to panel II in **i** and **j**).

the uppermost corundum sand marker that registers a total of 19.2 mm of subsidence (Fig. 3i, panel II).

### 3.1.2 Orthogonal rifting with syn-rift sedimentation – model A2

In the early rifting stages of model A2 ($t = 30$ min), strain analysis shows the deformation concentrating at the graben boundary faults (Fig. 3e). However, during these early rifting stages, the maximum normal strain values were lower inside the graben (Fig. 3e, f) than observed in model A1 (Fig. 3a, b). Towards the end of the model run, strain was homogeneously distributed between the boundary faults and the set of conju-

gate faults in the centre of the graben (Fig. 3f, j). Syn-rift sedimentation in model A2 (Fig. 3j) caused an increase in the graben width and subsidence compared to rifting without sedimentation in model A1 (Fig. 3i); the vertical offset of the first corundum sand marker shows a difference of ca. 1 cm between models A1 (19.2 mm; Fig. 3i, panel II) and A2 (30.9 mm; Fig. 3j, panel II), and the graben structure was ca. 1 cm wider in model A2 (65.2 mm) than in model A1 (56.2 mm).

## 3.2 Series B – inversion without syn-rift sedimentation

Here we show the results for the series B models that underwent two deformation phases (rifting and inversion) but without syn-rift sedimentation. We first present models B1 and B2 that involved orthogonal rifting, followed by models B3 and B4 with oblique rifting. These model pairs subsequently underwent either orthogonal or oblique inversion, respectively.

### 3.2.1 Orthogonal rifting followed by orthogonal (model B1) or oblique (model B2) inversion

The results from models B1 and B2 were very similar at the end of phase 1 and were also very similar to those of reference model A1 (Figs. 3a, b and 4a, b, i, j). Early rifting ($t = 30$ min; Fig. 4a and i) localized more strain along the graben normal faults than in the later rift phase ($t = 120$ min; Fig. 4b and j), since during the late rift stage, strain was distributed between the graben boundary faults and the intra-graben faults. Topography analysis (Fig. 4c, g, k, and o) shows a maximum graben subsidence of ca. 20 mm in both models B1 and B2.

After the first 60 min of orthogonal inversion, model B1 localized the strain both along the intra-graben faults and along new reverse faults on both sides of the graben (Fig. 4d). Towards the end of the model run, most parts of the southern reverse fault became relatively inactive, while the northern reverse fault grew and localized higher levels of strain (Fig. 4e). During the final stage ($t = 120$ min), the intra-graben faults had also become inactive (Fig. 4e). The areas immediately north and south of the graben were uplifted, while the floor of the inverted graben reached the same elevation as the pre-rift surface (Fig. 4f, h).

After the first 60 min of oblique inversion in model B2, strain was localized along the graben boundary faults (Fig. 4l), showing direct reactivation of the original graben faults only, which is in clear contrast to the orthogonal inversion of model B1 (Fig. 4d). At the end of phase 2, however, a single oblique reverse fault had appeared at the model surface grid, north of the graben, while all previous rift-related faults were inactive (Fig. 4m). The final topography data show a significantly higher maximum elevation than the pre-rift surface (15 mm difference) at the end of orthogonal inversion model B1 (Fig. 4f, h), while the oblique inversion model B2 (Fig. 4n, p) had ca. 7 mm higher maximum elevation than the pre-rift surface at the end of the model run.

### 3.2.2 Oblique rifting followed by orthogonal (model B3) or oblique (model B4) inversion

Oblique rifting ($\alpha = 45°$) of models B3 and B4 resulted in the development of two bands of en échelon normal faults bounding an E–W-striking graben after the first 30 min of deformation (Fig. 5a, i). At the end of phase 1, the strain results show that these en échelon faults had become interconnected, forming throughgoing, E–W-striking, and graben-bounding normal faults connected by oblique, WNW–ESE-trending lower-strain zones within the graben (Fig. 5b, j).

After 60 min of orthogonal inversion, model B3 showed the formation of a new straight reverse fault along the central axis of the graben and the development of a new reverse fault south of the graben (Fig. 5d, e). By the end of the inversion phase, after 120 min, the reverse fault remained active, while the fault in the centre of the graben became less so, with some parts being completely inactive (Fig. 5e). Uplift was more prominent in the area between the reverse fault and the graben, whereas in the northern part of the model a more widespread uplift was recorded (Fig. 5f, g).

After 60 min of oblique inversion in model B4, the oblique low-strain zones within the graben were partially reactivated, while a significant portion of the deformation localized along a new reverse fault to the north of the graben, and deformation started to localize in the south of the model as well (Fig. 5l). After 120 min of inversion, the northern reverse fault became almost completely inactive, and the deformation was fully localized on the southern reverse fault (Fig. 5m). The deformed surface grid registered the strike–slip component of the oblique movement along these reverse faults (Fig. 5m). Rift faults experienced only minor reactivation and became almost completely inactive by the end of the inversion phase (Fig. 5m; 120 min). The topography profiles indicate the uplift of the rift structures (17 mm elevation of the graben floor with respect to the depth of the initial graben floor at the end of rifting) and the new reverse faults on both sides of it (Fig. 5p), and while the northern reverse fault became inactive, distributed uplift affected the northern part of the model (Fig. 5p). Measured along the topographic profile, the maximum uplift away from the reverse faults was 5 mm in the north (where the reverse fault became inactive over time) and 2 mm in the south (where the reverse fault remained active).

## 3.3 Series C – inversion with syn-rift sedimentation

Here we present the results from our series C models with the rifting phase divided in eight sedimentation intervals of 15 min each, and with 20 mm h$^{-1}$ of displacement during both the rifting and subsequent convergence phases. The results, including sections, are presented in pairs according to the models' initial divergence direction (orthogonal and oblique, respectively; Figs. 6–8).

### 3.3.1 Orthogonal rifting with sedimentation followed by orthogonal (model C1) or oblique (model C2) inversion

The early stages of rifting of both models C1 and C2 resulted in high-strain localization at the graben boundary faults and lower-strain rates inside the graben (Fig. 6a, i). During later

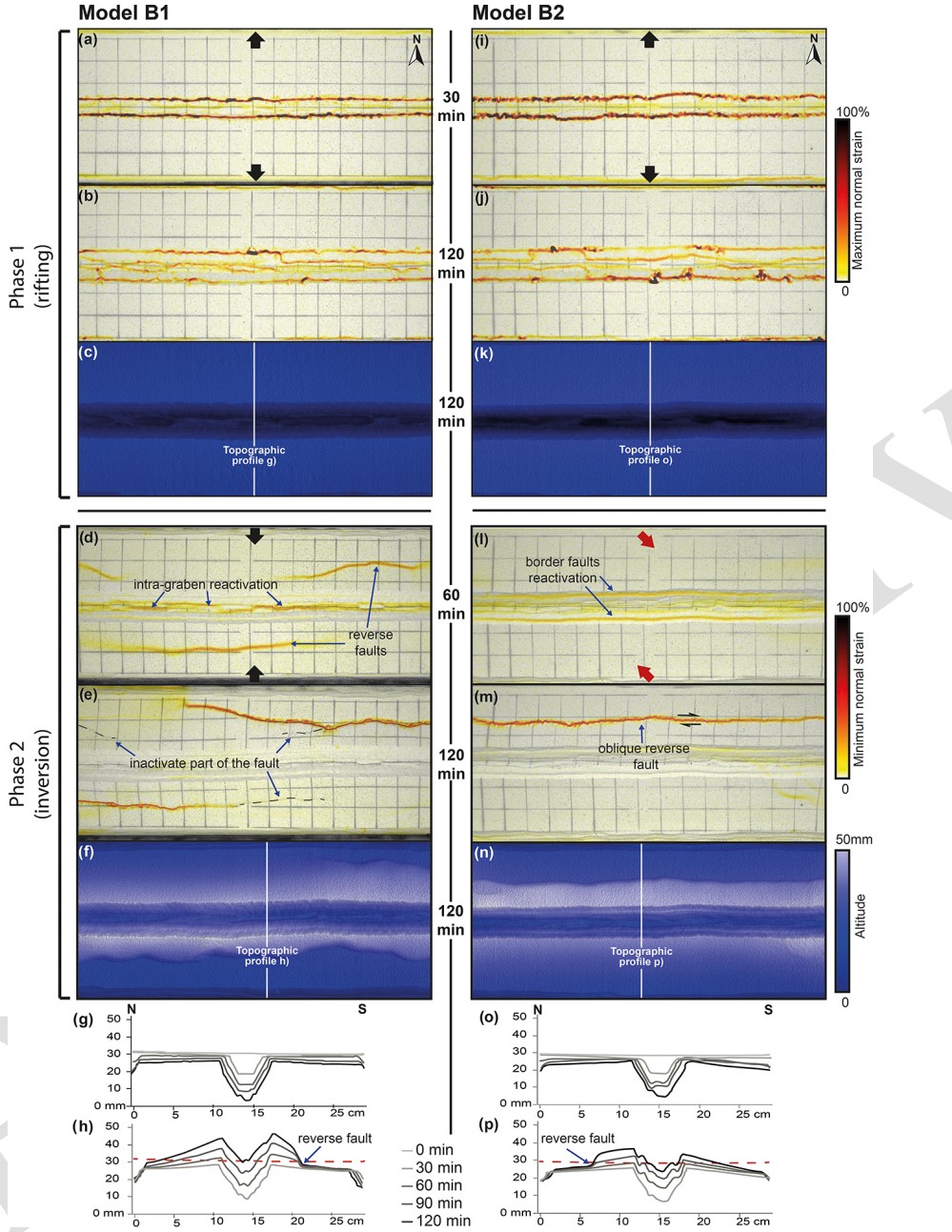

**Figure 4.** Evolution of deformation during rifting and inversion for models B1 and B2 (without sedimentation). **(a, b, i, j)** Top-view incremental maximum normal strain results for early- and late-stage rifting, respectively, projected on greyscale top view imagery of the model surface. **(c, k)** Digital elevation models at the end of rifting. **(d, e, l, m)** Top-view incremental minimum normal strain results for early- and late-stage inversion, respectively. **(f, n)** Top view of digital elevation model at the end of inversion. **(g, o)** Topographic profiles for every 30 min of rifting. **(h, p)** Topographic profiles for every 30 min of inversion. The vertical exaggeration is 4. The dashed horizontal red line indicates the initial surface level at the start of the model run.

rifting stages, the maximum normal strain values were lower along the graben boundary faults and instead rather evenly distributed among all faults within the graben (Fig. 6b, j). These results for the early and late stages of rifting show great similarity to the results from model A2 (Fig. 3e, f). Section thickness measurements from each of the 15 min syn-rift

sedimentation intervals (I1–8) indicate a progressive increase in the subsidence in the first two sedimentation intervals (up to 8 mm per interval; the inset in Fig. 7a, panel I; see I1–3). From interval I4 to I8, we observed a decrease in the subsidence rate (down to ca. 4 mm per interval; inset in Fig. 7a; see I4–8).

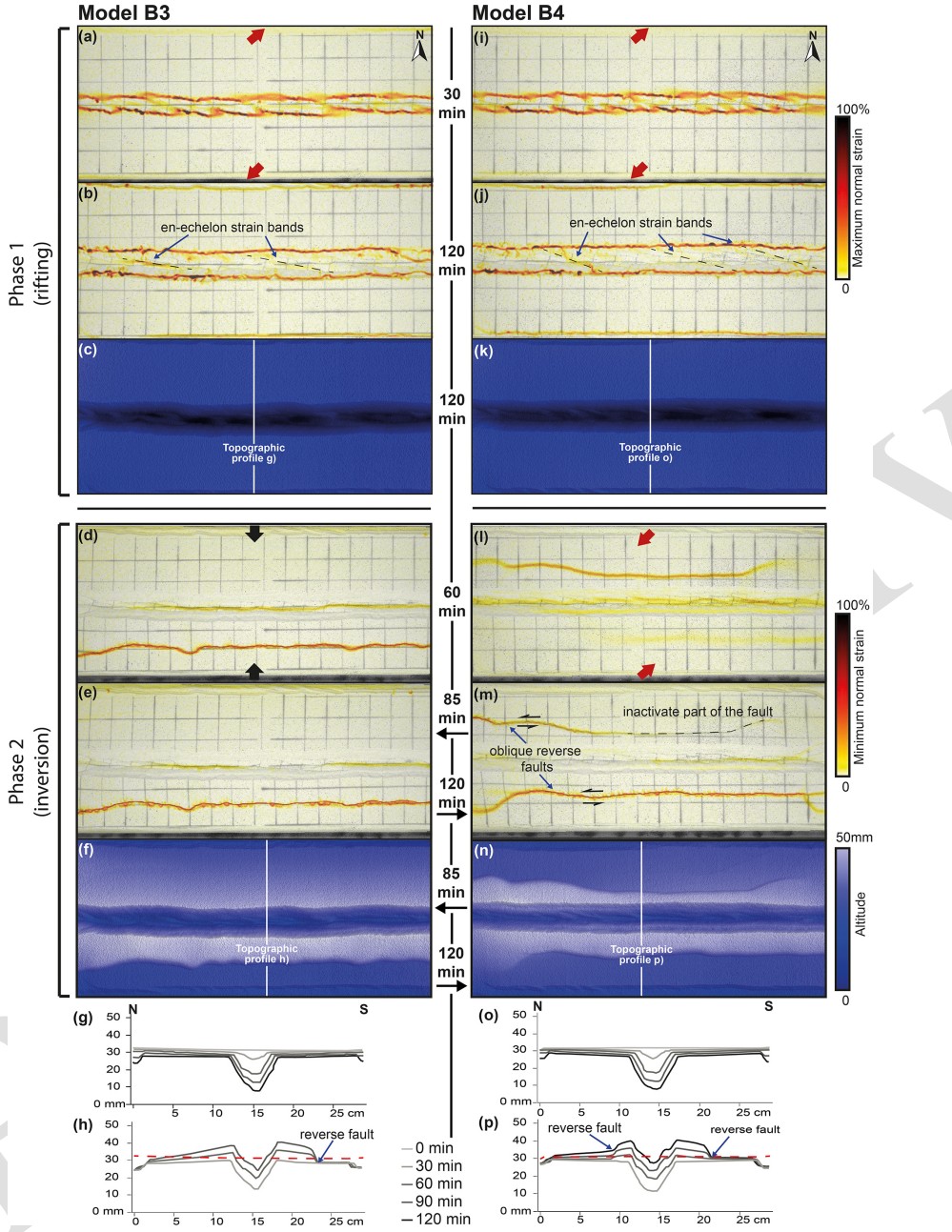

**Figure 5.** Evolution of deformation during rifting and inversion for models B3 and B4. **(a, b, i, j)** The top-view incremental maximum normal strain results for early- and late-stage rifting, respectively, projected on greyscale top-view imagery of the model surface. **(c, k)** Digital elevation models at the end of rifting. **(d, e, l, m)** Top-view incremental minimum normal strain results for early- and late-stage inversion, respectively, projected on greyscale top-view imagery of the model surface. **(f, n)** Top view of digital elevation model at the end of inversion. **(g, o)** Topographic profiles for every 30 min of rifting. **(h, p)** Topographic profiles for every 30 min of inversion. The vertical exaggeration is 4. The dashed horizontal red line indicates the initial surface level at the start of the model run. Note that model B3 had a reduced inversion duration of 85 min instead of 120 min, as indicated in the figure.

Orthogonal inversion in model C1 concentrated the deformation at a new reverse fault in the southern part of the model (Fig. 6d–e). Strain data show localization along this reverse fault, while no reactivation is visible along the inherited rift structures. In CE2 the section view (Fig. 7a), it becomes clear that the whole graben structure was uplifted by the reverse fault while the model surface was folded. The section shows that the reverse fault, in fact a ca. 1 cm thick shear zone by the end of the model run, was seeded in the viscous layer, which itself was also thickened (most probably already during rift-

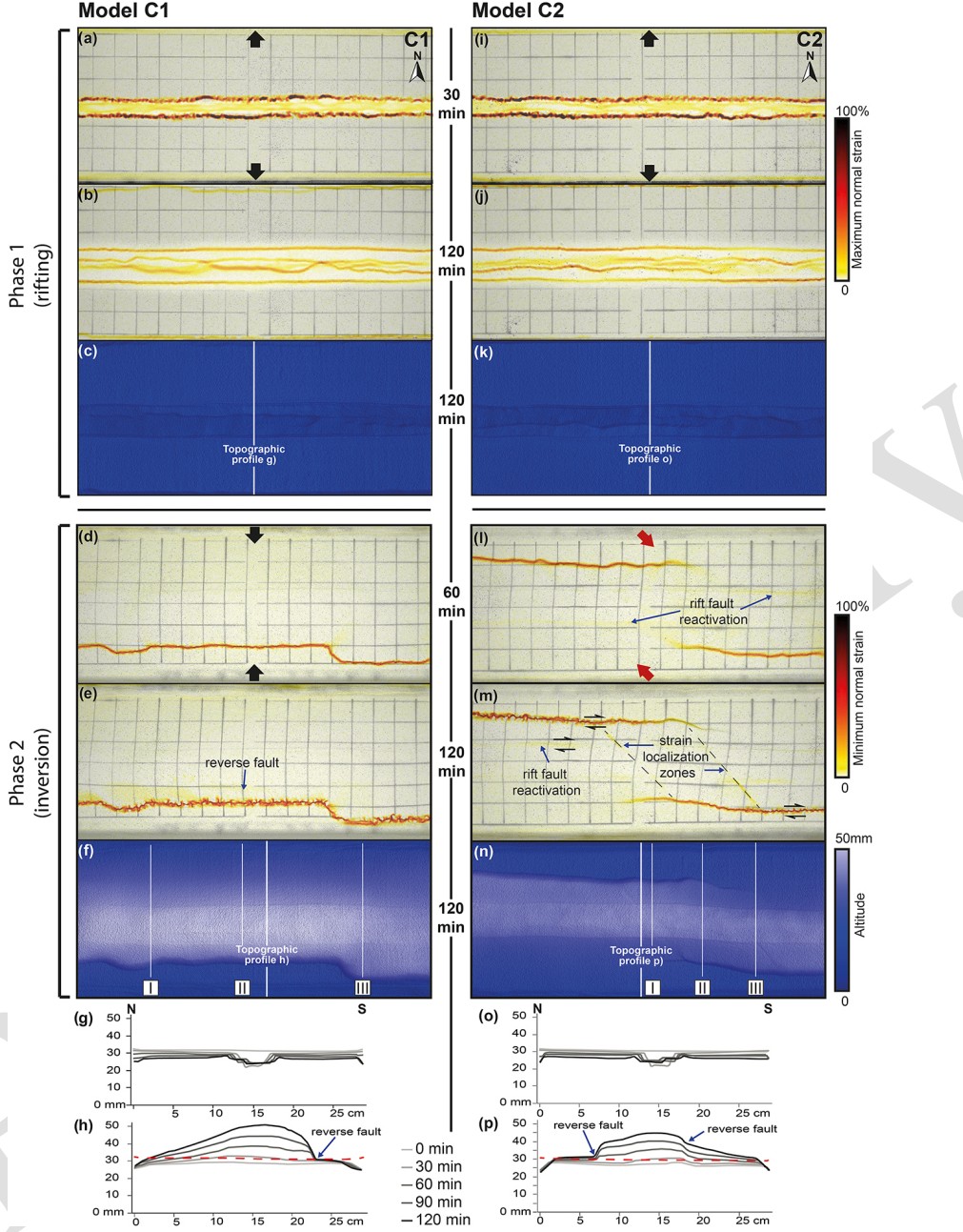

**Figure 6.** Evolution of deformation during rifting and inversion for models C1 and C2. **(a, b, i, j)** Top-view incremental maximum normal strain results for early- and late-stage rifting, respectively, projected on greyscale top-view imagery of the model surface. **(c, k)** Digital elevation models at the end of rifting. **(d, e, l, m)** Top-view incremental minimum normal strain results for early- and late-stage inversion, respectively, projected on greyscale top-view imagery of the model surface. **(f, n)** Top view of the digital elevation model at the end of inversion. **(g, o)** Topographic profiles for every 30 min of rifting. **(h, p)** Topographic profiles for every 30 min of inversion. The vertical exaggeration is 4. Topography is shown prior to syn-rift sedimentation for that interval, and the dashed horizontal red line indicates the initial surface level at the start of the model run.

ing as seen in sections from models A1 and A2; Figs. 3i, j, and 7a). We also note some apparent uplift along the graben border faults that are visible on topography data at the end of the model run but not significantly expressed on topography profiles (Fig. 6f, h). This is an artefact from the manual

addition of the graben infill during the rifting phase; the minimum normal strain results from our DIC analysis (Fig. 6d, e) show that the deformation was concentrated along the new reverse fault in the southern part of the model, whereas no

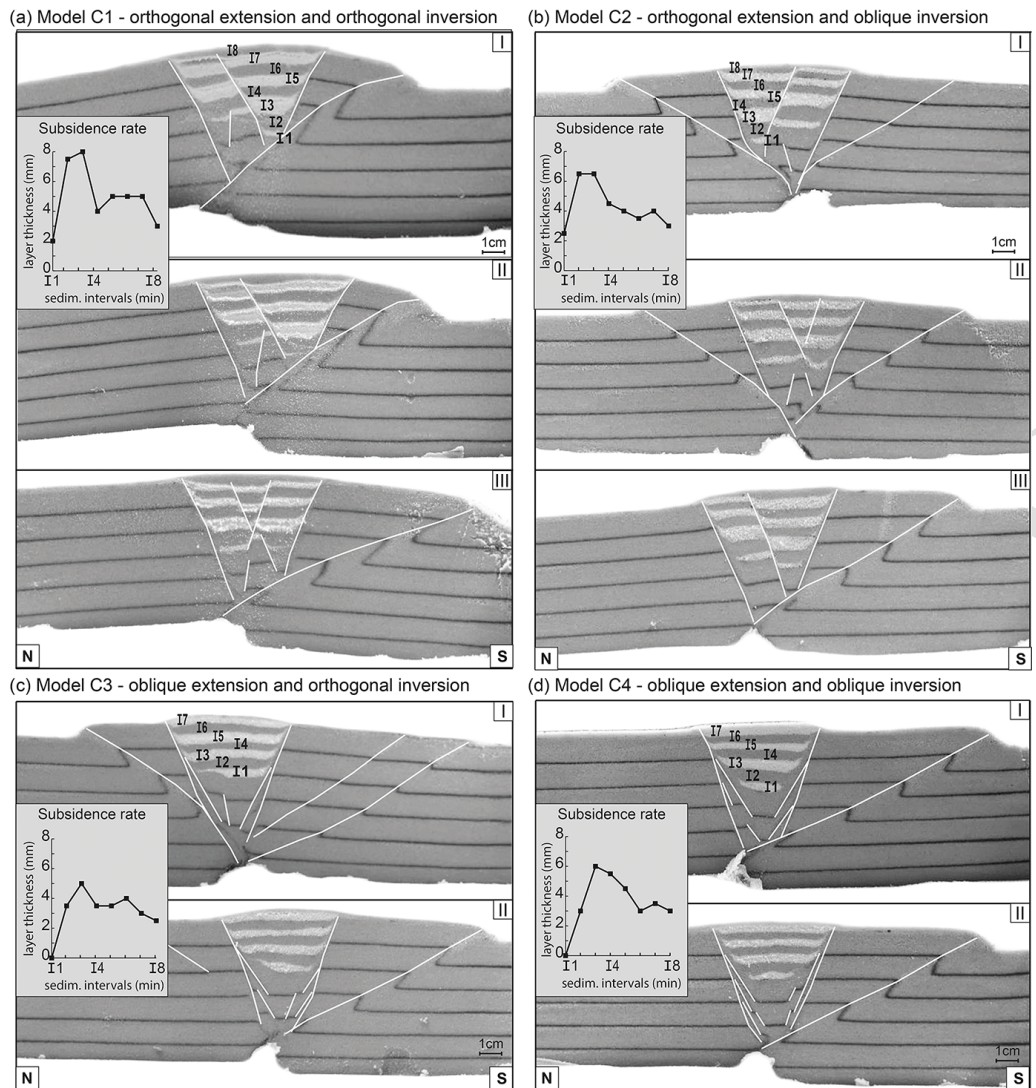

**Figure 7. (a–d)** Sections of experiments with sedimentation and measurements on models showing the influence of extension obliquity on sedimentation and subsidence rate. Section locations are shown in Figs. 5 and 8. Syn-rift sedimentation units always start with feldspar sand (white) and are divided into eight intervals of 15 min of extension, except for the oblique divergence models C3 and C4, where I1 and I2 are represented in the same unit. I1–8 indicate $t = 15, 30, 45, 60, 75, 90, 105$, and 120 min (after the initiation of rifting). The section orientations are indicated at the bottom section of each model.

discernable border fault reactivation was observed in model C1.

Compared to orthogonal inversion model C1, the oblique inversion in model C2 shows a different effect on the reactivation of previous rift structures (Fig. 6l–p). Strain data from our DIC analysis show a minor reactivation of the previously formed graben-bounding normal faults during the subsequent oblique inversion phase, while main strain localization was focused along newly formed reverse faults in the NW and SE quadrants connected by strain localization zones parallel to the inversion direction (Fig. 6l, m). Our topography analysis shows a small (ca. 2 mm) but distinct pop-up structure related to the minor reactivation of the graben border faults

(Fig. 6n, p), and the surface grid registered a small dextral strike–slip component of the reactivated border faults as well (Fig. 6m). In the section view, the newly formed reverse faults were shown to be, in fact, thick (ca. 1 cm) shear zones in those locations where only a single reverse fault developed, whereas the shear zones were thinner ($< 5$ mm) when multiple reverse faults developed (Fig. 7b).

### 3.3.2 Oblique rifting with sedimentation followed by orthogonal (model C3) or oblique (model C4) inversion

Models C3 (Fig. 8a) and C4 (Fig. 8i) developed clear en échelon graben boundary faults after the first 30 min of oblique

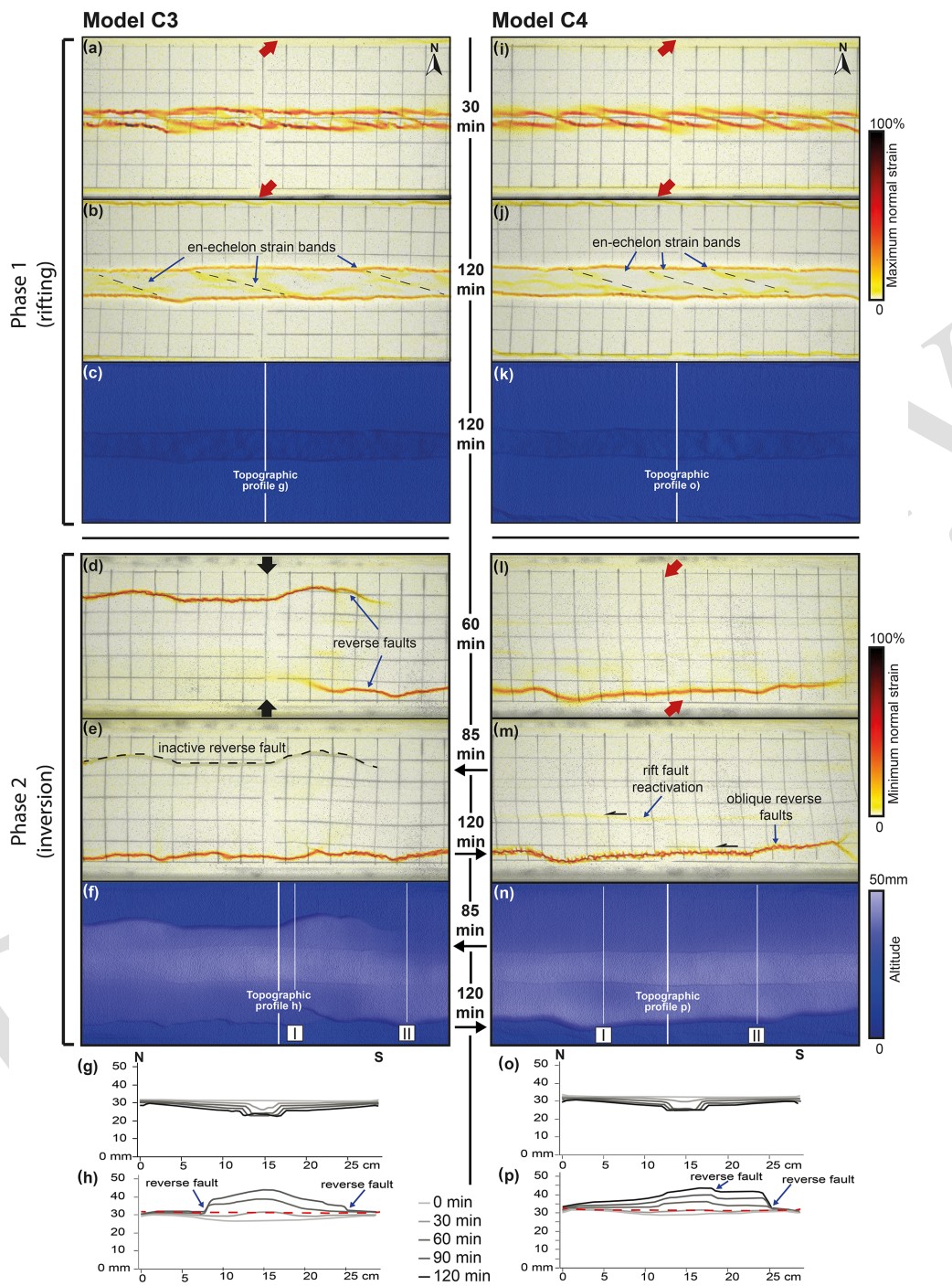

**Figure 8.** Evolution of deformation during rifting and inversion for models C3 and C4. **(a, b, i, j)** Top-view incremental maximum normal strain results for early- and late-stage rifting, respectively, projected on greyscale top-view imagery of the model surface. **(c, k)** Digital elevation models at the end of rifting. **(d, e, l, m)** Top-view incremental minimum normal strain results for early- and late-stage inversion, respectively, projected on greyscale top-view imagery of the model surface. **(f, n)** Top view of digital elevation model at the end of inversion. **(g, o)** Topographic profiles for every 30 min of rifting. **(h, p)** Topographic profiles for every 30 min of inversion. The vertical exaggeration is 4. Topography is shown prior to syn-rift sedimentation for that interval, and the dashed horizontal red line indicates the initial surface level at the start of the model run. Note that model C3 has a reduced inversion time of 85 min instead of 120 min, as indicated in the figure.

rifting, thus showing results similar to models B3 and B4 (Fig. 5b, i). Over the subsequent 1.5 h of rifting, the en échelon faults evolved into two main E–W graben boundary faults, but some faint late-stage en échelon strain bands remained active within the graben (Fig. 8b, j). Topography analysis showed that vertical subsidence in the first 30 min was lower than during the subsequent 30 min phases (2 mm per interval vs. 4.8 mm per interval; Fig. 7c, d). Subsidence was indeed slower in models C3 and C4 when compared to models C1 and C2; it took 30 min of oblique rifting (two 15 min intervals) to create accommodation space for sedimentation, while the first 15 min of orthogonal rifting in models C1 and C2 created enough subsidence to apply a sedimentation interval. Moreover, models C3 and C4 (Fig. 7c, d) did not develop the intra-graben normal faults seen in models C1 and C2 (Fig. 7a, b).

Orthogonal inversion in model C3 created initial reverse faulting in the north and SE of the models but without graben boundary fault reactivation (Fig. 8d). By the end of the experiment (Fig. 8e), after 85 min, the northern reverse fault became completely inactive, while the southern one grew laterally (westward), remaining active. Topography analysis shows uplift limited by the reverse faults on both sides of the model (Fig. 8f, h). In CE4 the section view, there is an along-strike switch between northern (Fig. 7c, panel I) and southern (Fig. 7c, panel II) reverse fault activity, and we also observe that reverse faults with larger offsets had an increased thickness. Similar to model C1, the apparent uplift along the graben border faults (Fig. 8f) is an artefact from the manual addition of graben infill during the rifting phase, since no discernable border fault reactivation appears on the inversion strain maps of model C3 (Fig. 8l, m).

The oblique inversion in model C4 is predominantly accommodated by a new reverse fault in the south, with limited reactivation of the rift structures as indicated by our DIC results (Fig. 8l–m). Furthermore, map view topography data indicate additional uplift in the initial graben (Fig. 8f). The topographic profiles (Fig. 8p) indicate a clear but limited reactivation of the graben boundary faults as well, starting after the first hour of the inversion phase and continuing until the end of the experiment.

The sections of models C3 and C4 (Fig. 7c, d) revealed that the reverse fault nucleated in the seed at the base of the graben and developed into a ca. 1 cm thick shear zone. Section II from orthogonal inversion model C3 (Fig. 7c) shows the presence of a reverse fault north of the graben that is seeding 2 cm below the surface, with no clear link to the previous rift faults or to the viscous material at the base of the graben, which is in contrast to the other reverse faults visible in Fig. 7. However, when assessing the model structures in map view (Fig. 8d–f), it becomes clear that this is in fact the tip of the same reverse fault present in section I of model C3 (Fig. 7c).

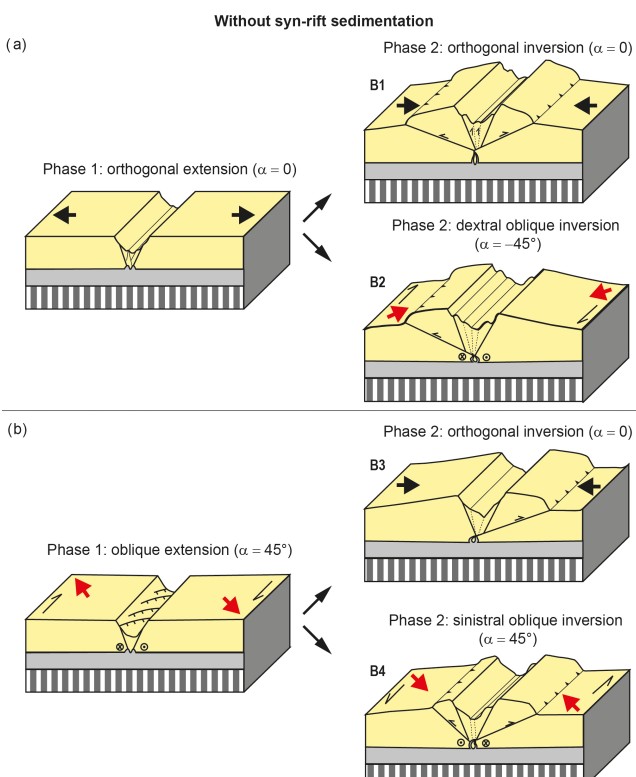

**Figure 9.** Schematic summary of our experimental results without syn-rift sedimentation.

## 4 Discussion

### 4.1 Summary and comparison to previous models

Our model results, presented in two schematic overview figures (Figs. 9 and 10), illustrate how imposed kinematics and the presence of syn-rift sedimentation affect the initial basin evolution and subsequent inversion.

### 4.1.1 Rifting phase

The overview of the rifting phase without sedimentation (Fig. 9) depicts the general differences in graben structure as a result of the divergence direction (orthogonal or oblique) in our models. A different divergence direction resulted in different initial graben structures. However, at the final stage of rifting, the graben geometries formed during orthogonal and oblique rifting were very similar (Fig. 9). The main difference occurred within the graben, where parallel pairs of conjugate normal faults formed due to orthogonal divergence, whereas oblique divergence resulted in en échelon fault structures. Furthermore, oblique divergence caused a decrease in graben width compared to the orthogonal rifting models, due to an increase in boundary fault dip, as also described in previous modelling studies (Tron and Brun, 1991; Zwaan et al., 2017, 2018a; Figs. 3 and 4). This reduction

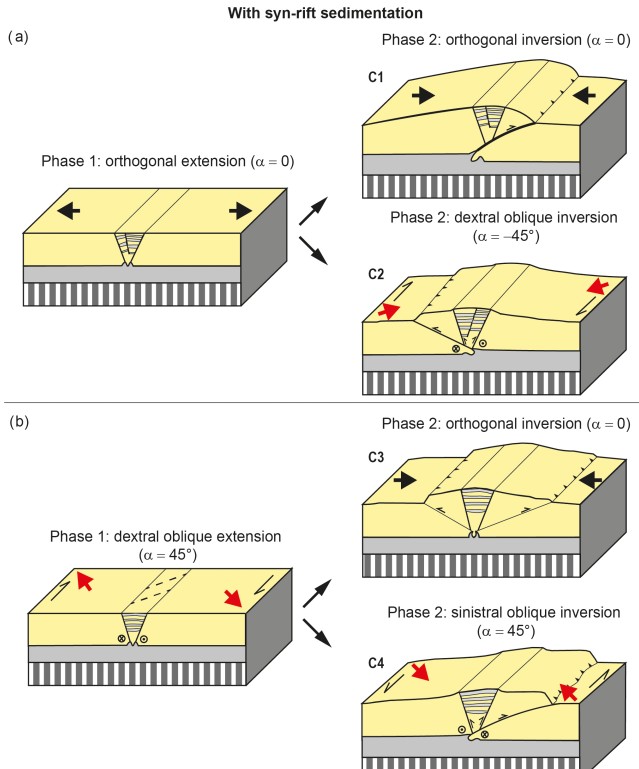

**Figure 10.** Schematic summary of our experimental results with syn-rift sedimentation.

in width and increase in fault angle is caused by the strike–slip component accommodating deformation in oblique rifting settings.

The syn-rift sedimentation models (Fig. 10) showed the same initial difference in orthogonal and oblique divergence as the models without sedimentation. The oblique divergence models resulted not only in a narrower graben at the end of the extension phase but also in a reduction in the final total subsidence observed in section (Fig. 7). A narrower graben forming during oblique rifting led to smaller sediment loads, and consequently, there was less graben floor subsidence. However, orthogonal and oblique rifting produced a very similar subsidence evolution in response to syn-rift sedimentation (Fig. 7). The first subsidence interval (I1) was always the smallest, while the subsequent three intervals (I2 to I4) accommodated more subsidence, and from this moment on, sedimentary intervals started thinning again until the last interval (I8). This initial subsidence rate increase likely occurred because the increase in sedimentary load over time enhanced subsidence. However, the reason why we observe a subsidence decline after sedimentation interval I4 remains unclear.

Overall, when assessing subsidence in models with and without syn-rift sedimentation, we observe that total subsidence in the former case was significantly higher while the

rift boundary faults remained active for a longer period of time as well. Zwaan et al. (2018a) reported a similar basin evolution due to syn-rift sedimentation. In their experiments without syn-rift sedimentation, the absence of sedimentary loading inside the graben led to a smaller offset along the graben boundary faults since part of the deformation was taken up by intra-graben faults. In contrast, in their models with syn-rift sedimentation, the graben wedge was strengthened so that faulting remained concentrated along the main graben boundary faults. The latter observation was also made in numerical models by Burov and Poliakov (2001) and Olive et al. (2014).

### 4.1.2 Inversion phase

Our experimental results have established an order of importance regarding the parameters controlling fault reactivation throughout the inversion phase (Figs. 9b and 10b). It seems that the rift kinematics, i.e. orthogonal vs. oblique rifting, have no significant influence on inversion structures, as the final rift structures were very similar; much more important are syn-rift sedimentation and inversion kinematics.

Without sedimentation, the rift structures were reactivated during inversion, and new low-angle reverse faults developed independently of the inversion direction (Fig. 9). Both orthogonal and oblique inversion resulted in the development of new low-angle reverse faults rooting at the base of the graben (Fig. 9). The reactivation of the rift structures occurred mainly at the intra graben structures in the orthogonal inversion models (Figs. 4 and 5; models B1 and B3), whereas in oblique inversion models (Figs. 4 and 5; models B2 and B4), both the graben boundary faults and the intra-graben faults showed significant reactivation.

The presence of syn-rift sediments (Fig. 10b) led to major differences in fault reactivation throughout the inversion phase, since the basin infill acted as a buffer to the reactivation of the rift structures. Our models results are in accordance with previous studies that described a similar decrease in fault reactivation when syn-rift sedimentation was applied (Pinto et al., 2010; del Ventisette et al., 2006; Panien et al., 2005; Dubois et al., 2002). In contrast, Panien et al. (2005) found that graben infill increased rift fault reactivation. This difference was likely due to their use of rheologically weak microbeads as graben infill, while we used feldspar and quartz sands so that the graben infill in our models had a similar rheology to the surrounding granular materials.

Furthermore, we found that during orthogonal inversion, graben faults did not undergo any reactivation, as the deformation localized in the newly formed low-angle reverse faults, whereas the limited reactivation of previous rift structures was observed in our oblique inversion models (Fig. 10). Other studies, with different analogue modelling set-ups, have also shown that increasing degrees of oblique convergence can promote normal fault reactivation (e.g. Nalpas et

al., 1995; Brun and Nalpas, 1996; see also the reviews by Bonini et al., 2012, and Zwaan et al., 2022b, and references therein). These observations are in line with earlier work by Sibson (1985, 1995), who demonstrated that relatively steep normal faults should not reactivate when put under orthogonal compressional stresses. Indeed, while analysing inverted rift basins in nature, Ziegler et al. (1995) found that in order to facilitate normal fault reactivation, the maximum horizontal compressive stress should be at an angle $< 45°$ to the normal fault strike.

## 4.2 Comparing model results with the Araripe Basin

This study was inspired by the Late Jurassic–Early Cretaceous Araripe Basin in NE Brazil, which is presently situated at 1000 m above sea level (Assine, 2007). This elevation is due to post-rift inversion for which two end-member scenarios have been proposed (regional uplift or rift fault reactivation; Peulvast and Bétard, 2015, and Marques et al., 2014, respectively; Fig. 1). Here we revisit these scenarios in the context of our model results, and propose a third, updated scenario for inversion in the Araripe Basin.

The uplift of the Araripe Basin infill, as explained by the Peulvast and Bétard (2015) scenario, involves a large-scale rather than local basin inversion produced by regional uplift (Fig. 1). According to these authors, the present-day high-standing mesa formation of the Araripe Basin is the result of differential erosion due to the presence of a strong sandstone formation covering the rift and post-rift sedimentary formations. However, other work demonstrates continuing ENE–WSW compression across the South American Plate (Assumpção, 1992; Coblentz and Richardson, 1996; Lima, 2003; Marques et al., 2013; Assumpção et al., 2016), combined with fault inversion in the region (e.g. Bezerra et al., 2020; Vasconcelos et al., 2021), suggesting that compressional horizontal stresses must have played a role in the inversion of the Araripe Basin as well.

Marques et al. (2014) proposed that inversion of the basin resulted from such regional horizontal compression acting on the South American Plate due to the opening of the South Atlantic Ocean to the east (ridge push) and the development of the Andes Cordillera to the west. Furthermore, Marques et al. (2014) concluded that these combined stresses were the cause for reactivation and inversion of high-angle normal faults. Additionally, the authors stated that the obliquity of the normal faults in relation to the inversion stresses, in combination with fluid injection along the fault planes, facilitated normal fault reactivation. However, although we observed some fault reactivation in our oblique inversion models, this reactivation never led to a full inversion of the graben normal faults (Figs. 9 and 10). In fact, no large-scale normal fault reactivation has been observed on seismic sections from the Araripe Basin either (Ponte and Ponte-Filho, 1996). Instead, Rosa et al. (2023) described limited reverse movement and fault inversion during Early Cretaceous rifting, when the basin changed from a system undergoing NE–SW extension to a system undergoing NW–SE extension. These authors reported positive flower structures on seismic lines that only affected syn-rift units and suggested that the inversion of normal faults, which Marques et al. (2014) attributed to the most recent inversion of the Araripe Basin, might in fact have occurred locally during the initial rifting phase instead. Furthermore, the post-rift sediments of the Araripe Basin cover an area larger than the extent of the original rift grabens and were deposited directly over the pre-Cambrian basement (Assine, 2007), and a large-scale offset of these post-rift units is not observed in the field.

However, recent work shows that mild post-rift fault inversion did take place in the Araripe Basin (Cardoso, 2010), and also other studies detected inversion in basins from the same rifting system that the Araripe Basin is part of (e.g. Rio do Peixe Basin and Potiguar Basin; Bezerra et al., 2020; Vasconcelos et al., 2021). These authors analysed seismic data and described a mild to moderate inversion along the normal faults of these basins, although no full-scale basin inversion, sensu Marques et al. (2014), was observed. Similar observations are made on seismic sections from the Araripe Basin (Ponte and Ponte-Filho, 1996), supporting the interpretation that horizontal shortening must have played a role in the inversion of the Araripe Basin. However, the bulk of this shortening must have been accommodated in some other way than large-scale normal fault inversion.

Our modelling results provide a possible solution to this apparent paradox, which involves the development of large-scale low-angle reverse faults during oblique convergence that take up most of the shortening, thus leading to basin uplift with some, but very limited, reactivation of the original rift structures (Fig. 11). Given the regional ENE–WSW shortening that is thought to have caused the inversion of the Araripe Basin and the NE–SW orientation of the initial grabens (Coblentz and Richardson, 1996; Marques et al., 2013; Fig. 1a), this oblique shortening was most likely of a dextral nature. Furthermore, the right-stepping en échelon arrangement of the Araripe Basin grabens is similar to oblique rifting structures in our models (Figs. 1a, 5, 8–10), possibly indicating an initial sinistral oblique rifting phase due to roughly E–W divergence, although this en échelon rift basin orientation may also have been influenced by the NE–SW-oriented shear zones found in the basement (Fig. 1a; de Matos, 1992; Ponte and Ponte-Filho, 1996). Our new oblique inversion scenario also explains the relatively undeformed uplift of the post-rift sediments and is in line with observations from the nearby Rio do Peixe Basin. In this basin, which is situated to the NE of the Araripe Basin and is part of the same rift trend, Vasconcelos et al. (2021) described mild to moderate inversion along the rift faults and reverse faulting in the basement outside the graben area. These observations of the Rio do Peixe Basin are in excellent agreement with our model results, and we propose that this same scenario can readily explain the structures observed in the Araripe Basin

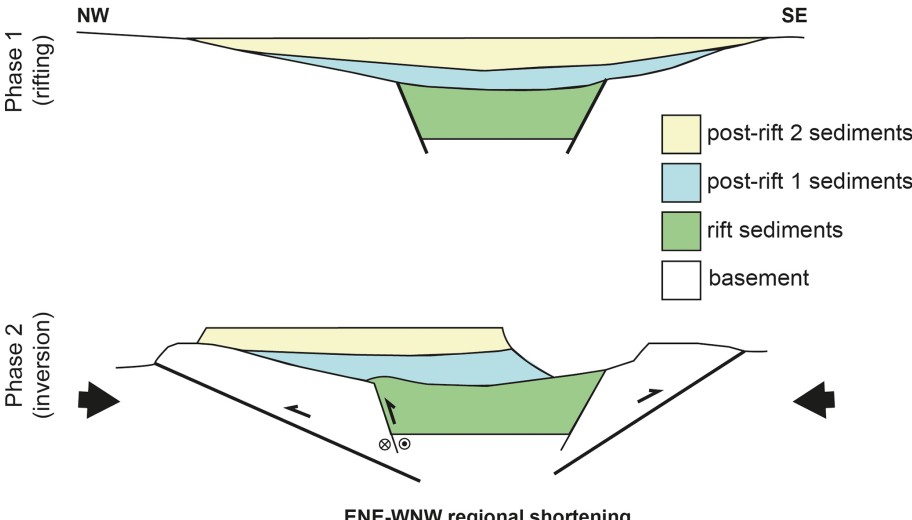

**Figure 11.** Proposed tectonic scenario for Araripe Basin inversion based on our analogue model results and data from the literature. The scenario involves an initial rifting phase creating SW–NE-oriented basins, followed by dextral oblique convergence due to general ENE–WSW-oriented convergence. See the text for details. This figure was modified after Marques et al. (2014).

as well (Ponte and Ponte-Filho, 1966; Cardoso, 2010; Rosa et al., 2023). In fact, Marques et al. (2014), who favoured large-scale reactivation of rift normal faults as the key inversion mechanism in the Araripe Basin, also reported the presence of some new reverse fault in the basement of the Araripe Basin area. Furthermore, the presence of large low-angle reverse faults (with a strike–slip component) outside the original rift basin in our models, combined with the observations from Marques et al. (2014) and other researchers discussed above, provides a strong incentive for further field investigations to verify our proposed scenario for inversion of the Araripe Basin.

## 5   Conclusions

In this study, we completed a series of new analogue modelling experiments aimed at evaluating the scenarios for basin inversion in the Araripe Basin in NE Brazil. We tested the influence of an orthogonal or oblique extension, followed by either orthogonal or oblique convergence on rift development and on subsequent inversion structures. We gain the following insights:

– During rifting without sedimentation, orthogonal divergence creates throughgoing border faults, whereas oblique divergence leads to the initial formation of en échelon faults that eventually will link up to establish large graben boundary faults. Rift basins with syn-rift sedimentation follow a similar evolution; however, the sedimentary loading increased subsidence when compared to models without sedimentation.

– During inversion, a major part of the deformation is accommodated by newly formed low-angle reverse faults. Within that framework, models without sedimentation saw significant intra-graben fault reactivation, which was roughly independent of the inversion direction (orthogonal or oblique). In contrast, in models with syn-rift sedimentation, the inversion caused only minor reactivation of the original graben boundary faults during oblique convergence, due to the sedimentary infill acting as a buffer. Orthogonal convergence in models with syn-rift sediments did not lead to rift fault reactivation.

– A comparison of the existing scenarios for inversion of the Araripe Basin with our model results and with data from the field shows that previous scenarios do not fully explain all observations of the natural example. Therefore, we propose an alternative scenario based on our model results, which involve dextral oblique inversion and the development of low-angle reverse faults (with a strike–slip component) outside the basin. This scenario provides an incentive for future (field) studies in the Araripe Basin area.

*Data availability.*   Detailed overviews of model results are publicly available in the form of a data publication available from the GFZ Data Services database (Richetti et al., 2023), which can be accessed at https://doi.org/10.5880/GFZ.2.5.2023.003.

*Author contributions.*   PCR, FZ, GS, and RSS planned and designed the experiments. PCR completed the experiments, analysed the model results, and wrote the first draft. FZ participated in running some of the experiments, and FZ and TCS helped in perform-

ing the model analysis. PCR, FZ, GS, RSS, and TCS participated in the interpretation of the model results and reviewed and edited the paper.

*Competing interests.* The contact author has declared that none of the authors has any competing interests.

*Special issue statement.* This article is part of the special issue "Analogue modelling of basin inversion". It is not associated with a conference.

*Acknowledgements.* Pâmela C. Richetti and Renata S. Schmitt gratefully acknowledge the support from research and development project "Correlação estratigráfica, evolução paleoambiental e paleogeográfica e perspectivas exploratórias do Andar Alagoas", sponsored by Shell Brasil Petróleo Ltda, and the strategic importance of the support given by ANP (Brazil's national oil, natural gas, and biofuels agency) through the R&D levy regulation (technical cooperation no. 20.219-2). Pâmela C. Richetti acknowledges Coordenação de Aperfeiçoamento de Pessoal de Nível Superior (grant no. 88887.569825/2020-00) – for their financial support. Frank Zwaan and Timothy C. Schmid have been funded by the Swiss National Science Foundation (grant no. 200021-178731, https://data.snf.ch/grants/grant/178731, last access: 1 October 2023, awarded to Guido Schreurs), which also covered the open-access publication costs. Frank Zwaan has also been funded by a GFZ Discovery Fund fellowship. Renata S. Schmitt acknowledges funding from the Conselho Nacional de Desenvolvimento Científico e Tecnológico (grant no. 311748/2018-0), Fundação Carlos Chagas Filho de Amparo à Pesquisa do Estado do Rio de Janeiro (grant no. E-26/200.995/2021), and Swiss National Science Foundation (grant no. IZSEZ0_191196/1, https://data.snf.ch/grants/grant/191196, last access: 1 October 2023) research grants. We thank Florian Ott and Kirsten Elger for helping us creating a GFZ data publication containing supplementary material (Richetti et al., 2023), reviewers Fernando Ornelas Marques and Ioan Munteanu for their constructive feedback, and editor Ernst Willingshofer for handling the review process.

*Financial support.* This research has been supported by the Coordenação de Aperfeiçoamento de Pessoal de Nível Superior (grant no. 88887.569825/2020-00), the Swiss National Science Foundation (grant nos. 200021-178731 and IZSEZ0_191196/1), the Conselho Nacional de Desenvolvimento Científico e Tecnológico (grant no. 311748/2018-0), the Fundação Carlos Chagas Filho de Amparo à Pesquisa do Estado do Rio de Janeiro (grant no. E-26/200.995/2021), Shell Brasil Petróleo Ltda (project: Correlação estratigráfica, evolução paleoambiental e paleogeográfica e perspectivas exploratórias do Andar Alagoas), ANP (technical cooperation no. 20.219-2), and the GFZ Discovery Fund.

*Review statement.* This paper was edited by Ernst Willingshofer and reviewed by Fernando Ornelas Marques and Ioan Munteanu.

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

**Remarks from the language copy-editor**

CE1  "View" requires an article. An exception would be if "section view" refers to a mode of viewing (e.g., when viewing in section view).

CE2  See comment above.

CE3  See comment above.

CE4  See comment above.

**Remarks from the typesetter**

TS1  Concerning the placement of figures. Please note that for the HTML view of your paper, the figures and tables have been placed where they are first mentioned in the text. Due to the fact that they can only be placed at the top of a page in the PDF version, figures and tables might not be displayed exactly where they are mentioned in the PDF. Please keep in mind that the figures and tables might not be displayed where they are mentioned in the HTML version.

TS2  This correction needs an approval from the editor.