# Peer review of "Analogue modelling of basin inversion: the role of oblique kinematics and implications for the Araripe Basin (Brazil)"

_EGUsphere, 2022_

## Referee Comment (RC1)

[referee-annotated manuscript omitted]

---

## Referee Comment (RC2)

[referee-annotated manuscript omitted]

---

## Author Comment (AC1)

**Referee 1 - Fernando Ornelas Marques**

*Assessment:* the topic of the ms. is relevant for geosciences, and therefore suitable for EGUsphere, but not in its present form. The ms. needs major revision before matching the high standards of the journal. Based on their experimental results, the authors conclude that the scenario proposed by Marques et al. (2014) for the inversion of the Araripe Basin is not viable. This is wrong because they did not test the arguments used by Marques et al. (2014), which are much lower angle between shortening direction and graben strike (<45º), and fault lubrication by injected soft clays. Therefore, all the authors may conclude is that 45º are not enough to explain the amount of inversion in the Araripe Basin. This is the main problem that the authors have to solve. The authors should read more carefully what previous authors have said about the mechanics of inversion of normal faults (e.g. Sibson 1985; Brun and Nalpas, 1996; Marques and Nogueira, 2008), in particular what Marques et al. (2014) proposed for the Araripe Basin.

- **Answer:** Thank you for your review, it is very important to have a review from an author that worked in the area of interest. First of all, we agree that we have to discuss more papers on the inversion of the Araripe. We have improved that in the revised version and added more references to these papers.

- Regarding the reactivation of rift faults, we must unfortunately disagree with the reviewer. The main rift fault system is oriented NE-SW to ENE-WSW. If we look at the map from Marques et al. (2014), the indicated reverse faults are mainly oriented NW-SE, therefore these faults are not related to the pre-existing rift system (NE-SW).

- In our figure 1, we added the main NE-SW normal faults that controlled the Araripe rift phase. This figure was drawn using existing geological and aeromagnetic maps (Scherer et al., 2014; Camacho and Souza, 2017), and seismic interpretations (Rosa et al., 2022). The grabens shown on the figure are well-known from the literature.

- Moreover, the argument that fluid-assisted deformation could assist inversion of faults may on itself be valid, but we argue that these NW-SE faults do not represent the main rift system and there is no evidence, in seismic studies (Ponte and Ponte-filho, 1996, Rosa et al., 2022), for large-scale fault inversion due to

horizontal compression anywhere in the Araripe Basin. Therefore, this argument, like the argument of more oblique convergence to explain the inversion of the Araripe Basin rift structures, is superfluous.

- Instead, it may be that Marques et al. (2014) over-extrapolated the structures they found in the field. A recent study by Rosa et al. (2023) suggested the existence of two phases of extension with different divergence directions during rifting in the Araripe Basin. Such a reorientation of divergence may have caused some minor reactivation and inversion of the normal faults described by Marques et al. (2014).

- However, there are in fact some signs of inversion on seismic sections from the Araripe Basin, as well as from other basins in the region (Rio do Peixe Basin, Vasconcelos et al., 2021). It is just by far not as intense as the Marques et al. (2014) scenario. As such, we still need to explain where the deformation was expressed in the Araripe Basin since it did not cause the inversion of the original rift structures, and here our analogue models come into play.

- Our models show that for various scenarios, the bulk of inversion can be accommodated by new reverse faults. This fits very nicely with the findings from the recent work by Vasconcelos et al (2021) on the nearby Rio do Peixe Basin, where such reverse faults were found outside of the main basin.

- We now added this information in the introduction, and discussion parts of the text, presenting a somewhat revised argument* that still leads to the same interpretation as in the previous version: we predict that the bulk of inversion in the Araripe Basin is accommodated by new reverse faults.

* Revised argument in the new manuscript:

- there are two end-member models for the Araripe Basin uplift: regional uplift (Peulvast and Bétard, 2015) and full inversion of the old rift structure (Marques et al. 2014)

- We know that inversion happened (minor inversion seen on seismic lines, indications from other basins) so that the Peulvast and Bétard (2015) model is not complete.

- But all available data indicates that it did not cause the large-scale inversion of the rift basin structures that Marques et al. (2014) proposed.

- Thus, we ran a series of analogue models to explore how inversion of the basin could have taken place.

- We find that new reverse faults can be a good explanation as to how inversion in the Araripe Basin may have occurred, which is in line with observations from nearby inverted basins.

**Main comments**

Models with orthogonal and oblique inversion cannot be directly compared because the amount of extension (rift phase) and shortening (inversion phase) are not the same (smaller in the oblique inversion). This is because the run time is the same for most experiments, and even worse when the inversion time was reduced from 120 to 85 minutes. It is easy to see the problem using vectors and simple trigonometry. Angle of 45º for the inversion phase – Brun and Nalpas (1996) showed experimentally that the angle between graben strike and shortening direction must be < 45º for inversion of precursor normal faults to take place. They also show in their Fig. 4 that at 45º new thrusts form, and that inversion of normal faults is minimal, similarly to the experiments presented by Richetti et al.. Therefore, what these authors are showing is that 45º is too much, and so they cannot argue that reactivation of precursor normal faults is not enough to explain the Araripe inversion. Make your definition of angle alfa equal to Brun and Nalpas' definition for consistency. For the non-expert reader it becomes confusing, because your alfa is the complementary angle of Brun and Nalpas' definition.

- **Answer:** We agree that there may be some issues if one would attempt a quantitative comparison. However, we are not doing so, as we are more interested in the general behaviour of the system by means of a qualitative comparison. The differences in main structures are quite clear when comparing our various models. As such, we believe that not having the exact same amount of shortening in a couple of our models is not a big issue.

- Please note that different analogue modellers use different definitions of angle alpha in their papers (either the angle between the normal to the rift axis and the displacement direction, or the angle between the rift axis and the displacement directions). We believe that our definition (the latter) is more intuitive as it

means that orthogonal divergence or convergence is defined as alpha = 0° (no obliquity) and prefer to keep it as is.

Richetti et al. say in lines 497-499, and I quote: "*However, although we observed some fault reactivation in our oblique inversion models, this reactivation did never lead to full inversion of the graben normal faults (Figs. 9 and 10), which* **contradicts** *the Marques et al. (2014) scenario*". **No, it does not contradict**. We proposed a much lower angle between shortening direction and graben strike (you can check in Fig. 6B). Besides, we also considered fault weakening as a mechanism that can promote inversion (read text upfront in the Abstract, and look at Fig. 11 for a field example) as experimentally shown by Marques and Nogueira (2008), which you should cite when discussing mechanisms of normal fault inversion and the Araripe Basin.

- **Answer:** We deleted part of the sentence the reviewer does not agree with, but no indications of large-scale inversion is observed in other (field and geophysical) studies of the Araripe Basin and other basins in the region. Therefore, the discussion whether higher degrees of oblique convergence, or fault weakening did occur, is in fact irrelevant. As described above, we have rewritten part of the manuscript to better reflect this argument.

Richetti et al. further say in lines 514-515, and I quote: "*We thus find that neither of the two end-member scenarios seems to fully explain the inversion observed in the Araripe Basin area.*". This is simply wrong, for two reasons: (1) you did not test Peulvast and Bétard's hypothesis; (2) you did not test what Marques et al. (2014) proposed for the Araripe inversion, which is low inversion angle and fault lubrication.

- **Answer:** We disagree, as pointed out in the new manuscript, both end-members do not work: we see that there is some localized inversion going on in the area, but we do not see the large-scale inversion of rift faults as proposed by Marques et al. (2014). However, field data from other basins nearby does show the kind of reverse faults away from the basin that we would expect based on our modelling results.

Fault lubrication – Marques et al. (2014) proposed that inversion was facilitated by injection of soft materials (mostly clay, but most probably also fluid overpressure; e.g.

Cobbold and Castro, 1999; Mourgues and Cobbold, 2003) into the precursor normal faults. This effect was shown experimentally by Marques and Nogueira (2008), who concluded that normal fault inversion, even by orthogonal compression, is possible if, and only if, the fault friction is greatly decreased. Given that Richetti et al. did not test the effects of fault lubrication, they should be more cautious when discussing what Marques et al. (2014) said about the inversion of the Araripe Basin, and they should cite Marques and Nogueira (2008) to support what Marques et al. (2014) proposed.

- **Answer:** We simply don't see the large-scale fault reactivation in any other studies that show the Araripe basin seismic lines, so the argument regarding the effects of fault weakening is irrelevant (see previous comments on this topic).

**Abstract**

**Line 14**

- **Comment:** infill currently found
- **Answer:** Thanks for the suggestion, it is modified.

**Line 14**

- **Comment:** "is proposed" gives the impression to the reader that the idea is yours, which is not the case. Therefore, it should be replaced by "has been proposed by previous authors"
- **Answer:** Thanks for the suggestion, it is modified.

**Line 17**

- **Comment:** This is not correct, you only tested one scenario, the tectonic inversion scenario
- **Answer?**

**Line 23**

- **Comment:** Échelon

- **Answer:** Thanks for the suggestion, it is modified.

**Line 25**

- **Comment:** What angle between shortening direction and master border fault strike?
- **Answer:** The angles between shortening directions and master faults are 45° and 90°.

**Line 30**

- **Comment:** This is not true. Marques et al. (2014) include all your experimental results, mainly new reverse faults and inverted normal faults. Additionally, Marques et al. (2014) also explain normal fault inversion by weak fault rock that lubeicates the fault during inversion.
- **Answer:** The rift faults should be reactivated and highly inverted in our experiments to include all Marques et al. (2014) results. See previous comments on why fault lubrification/weakening (nor highly oblique convergence) does not solve the issues of the missing large-scale inversion of original rift faults.

**Line 31**

- **Comment:** I do not see why this an alternative to the explanation given By Marques et al. (2014)
- **Answer:** It is different since we propose no major reactivation of rift boundary faults. Instead, we propose the formation of new reverse faults away from the original basin.

**1 Introduction**

**Line 48**

- **Comment:** Sinistral

- **Answer:** Thanks for the suggestion, it is modified.

**Line 52**

- **Comment:** and ca. 500 m above the surrounding basement
- **Answer:** Thanks for the suggestion, it is modified.

**Line 54**

- **Comment:** You have to cite Gurgel et al. (2013), Nogueira et al. (2015) and Ramos et al. (2022)
- **Answer:** Thanks for the suggestion, we added the references.

**Line 55**

- **Comment:** maximum compressive stress
- **Answer:** Thanks for the suggestion, it is modified.

**Line 60**

*"According to Marques et al. (2014), this compression caused the complete inversion of the initial high angle normal faults of the Araripe Basin (Fig. 1e) through an oblique compression and injection of soft material into these faults."*

- **Comment:** and the creation of new low angle reverse faults in
- **Answer:** Marques et al (2014) figure of their inversion model does not represent new low angle reverse faults in the basement outside the rift grabens. That's very different from our results; the new reverse faults are very important in our models.

**Line 61**

- **Comment:** into
- **Answer:** Thanks for the suggestion, it is modified.

**Line 66**

- **Comment:** Brun and Nalpas (1996) must be cited here
- **Answer:** Thanks for the suggestion, added the reference.

**Line 67**

- **Comment:** Marques and Nogueira (2008) should be included here for inversion, because the precursor normal faults have been more easily inverted due to weakening of the fault rock, which decreases friction
- **Answer:** Thanks for the suggestion, added the reference, but as stated earlier, it seems that fault weakening did not play an important role as rift faults did not experience major inversion.

**Line 71**

- **Comment:** and differential erosion (basin sediments more resistant to erosion than basement granitic and metamorphic rocks)
- **Answer:** Thanks for the suggestion, it is modified.

**Line 75**

- **Comment:** tectonic inversion
- **Answer:** Thanks for the suggestion, it is modified.

**Line 75**

- **Comment:** These authors do not consider tectonic inversion a viable mechanism. They say it up front in the Abstract
- **Answer:** Thanks for the suggestion, it is modified.

**Line 76**

- **Comment:** could have taken
- **Answer:** Thanks for the suggestion, it is modified.

**Line 77 (figure 1)**

- **Comment:** These normal faults do not agree with inversion as proposed by Marques et al. (2014). Check Fig. 6b of Marques et al. (2014)
- **Answer:** This is because we compiled the rift faults from other works (these are cited in the figure caption), since inversion structures in Marques et al. (2014) are mainly oriented NW-SE and the faults are no plotted over a map, while the DEM image cuts out the main east portion of the basin where most rift faults are found. In conclusion, Marques et al. (2014)'s figure 6b proposes new inversion faults (which, as pointed out earlier, are not identified on seismic lines) and no rift inverted rift-related faults.

**2 Methods**

**2.1 Model set up**

**Line 90**

- **Comment:** end walls, because it is confusing to have 4 sidewalls
- **Answer:** Thanks for the suggestion, it is modified.

**Line 91**

- **Comment:** Thick
- **Answer:** Thanks for the suggestion, it is modified.

**Line 92**

- **Comment:** Intercalated
- **Answer:** Thanks for the suggestion, it is modified.

**2.2 Materials**

**Line 133**

- **Comment:** quartz
- **Answer:** Thanks for the suggestion, it is modified.

**2.3 Model parameters**

**Line 149**

- **Comment:** All this is a major problem, because you cannot compare final results of orthogonal and oblique rifting and inversion

- **Answer:** The comparison is indeed not 100%, as the timing is not exactly the same, but still we can do a highly useful comparison as the general structural template is established early on. Longer experimental duration does not significantly change the major features.

**Line 157 (table 2)**

- **Comment:** Why not?!
- **Answer:** We do not show these sections because we only made them for one model in series B, and we do not focus on these models as there is no sedimentation in these models (therefore they are less realistic). We now added in the table that these cross-sections are presented in the supplementary material.

**2.4 Scaling**

**Line 163**

- **Comment:** Nature
- **Answer:** Thanks for the suggestion, it is modified.

**3 Results**

**Line 229**

- **Comment:** This figure does not exist in the PDF I received
- **Answer:** The sections are included in the supplementary material and we corrected the text.

**3.1 Series A – Reference models**

**Line 233**

- **Comment:** delete the hyphen
- **Answer:** Thanks for the suggestion, it is modified.

**3.1.1 Orthogonal rift without syn-rift sedimentation - Model A1**

**Line 237**

- **Comment:** With
- **Answer:** Thanks for the suggestion, it is modified.

**Line 242**

- **Comment:** Shows
- **Answer:** Thanks for the suggestion, it is modified.

**Line 242**

- **Comment:** associated with
- **Answer:** Thanks for the suggestion, it is modified.

**Line 246**

- **Comment:** of the rifting
- **Answer:** Thanks for the suggestion, it is modified.

**Line 247**

- **Comment:** two master faults bounding the
- **Answer:** Thanks for the suggestion, it is modified.

**Line 248**

- **Comment:** subsidence
- **Answer:** Thanks for the suggestion, it is modified.

**3.1.2 Orthogonal rifting with syn-rift sedimentation – Model A2**

**Line 253**

- **Comment:** At what stage?
- **Answer:** Thanks for the suggestion, we added details to the text.

**Line 258**

- **Comment:** Models
- **Answer:** Thanks for the suggestion, it is modified.

**Line 260 (figure 3)**

- **Comment:** You must give the used vertical exaggeration in d and h
- **Answer:** Thanks for the suggestion, we added to the figure caption.

**Line 260 (figure 3)**

- **Comment:** There is something wrong when comparing d and h with i and j, because the width of the final graben in h is significantly smaller than in d, which is the opposite of i and j

- **Answer:** We do not really follow what the issue is here. The graben is correctly depicted between h and j. Perhaps the issue is that the topographic profiles overlap due to the constant sedimentary infill?

**Line 260 (figure 3)**

- **Comment:** Vertical axes are missing. The reader needs dimensions

- **Answer:** There must be some confusion, as the vertical displacement measurements are there in the figure.

**Line 261**

- **Comment:** Figure caption

- **Answer:** Thanks for the suggestion, it is modified.

**3.2 Series B – inversion without sedimentation**

**Line 271**

- **Comment:** models

- **Answer:** Thanks for the suggestion, it is modified.

**Line 272**

- **Comment:** models

- **Answer:** Thanks for the suggestion, it is modified.

**3.2.1 Orthogonal rifting - orthogonal ( B1) and oblique inversion (B2)**

**Line 275**

- **Comment:** oblique (B2)
- **Answer:** Thanks for the suggestion, it is modified.

**Line 276**

- **Comment:** This is a repetition of A1. Delete
- **Answer:** I described model B1 rifting phase in this sentence I can't delete it.

**Line 281**

- **Comment:** What is initially? How many minutes?
- **Answer:** Thanks for the suggestion, it is modified.

**Line 284**

- **Comment:** Give time (minutes) to all these stages that you describe
- **Answer:** Thanks for the suggestion, it is modified.

**Line 285**

- **Comment:** adjacent to
- **Answer:** Thanks for the suggestion, it is modified.

**Line 285**

- **Comment:** Of
- **Answer:** Thanks for the suggestion, it is modified.

**Line 287**

*"After the first hour of oblique inversion in Model B2, strain was localized along the graben border faults (Fig. 4l) showing direct reactivation of the original graben faults only, in clear contrast to the orthogonal inversion of Model B1 (Fig. 4d). At the end of Phase 2, however, a single oblique reverse fault had appeared at the model surface grid, north of the graben, while all previous rift related faults were inactive (Fig. 4m). The final topography shows a significantly higher maximum elevation than the pre-rift surface of ~15 mm in orthogonal inversion Model B1 (Fig. 4f, h), while the oblique inversion Model B2 (Fig. 4n, p) had an ~7 mm higher elevation than the pre-rift surface."*

- **Comment:** You should quantify all this description by making measurements on the topographic profiles and produce graphs with evolution over time

- **Answer:** The evolution over time of the topography is already provided. We already added some quantification and are not sure what further quantification is requested here.

- **Comment:** You should also draw on the profiles the inversion faults within the graben

- **Answer:** They are provided within the cross-sections, wherever available. For models without cross-sections, it is not possible to place these faults with confidence. However, the new reverse faults can be indicated over the topography profile, and we already indicated them.

**Line 294 (figure 4)**

- **Comment:** Insert dashed lines in d, e, l and m that represent the master border faults at the end of the rifting phase, so that we can better visualize the effects of shortening

- **Answer:** We opted not to insert more dashed lines in these pictures since it would make it too crowded and we can see the faults clearly as they are.

- **Comment:** Minimum
- **Answer:** Thanks for the suggestion, it is modified.

- **Comment:** Zero should be the initial horizontal topographic surface
- **Answer:** Thanks for the suggestions, we made this change in the figures

- **Comment:** Why are g and o so different?
- **Answer:** They are not that different in fact, the only difference is the graben shape and not the general subsidence pattern. We believe is it due to sand collapse during the rifting process. Some variation is to be expected in analogue models, this is not an issue here.

- **Comment:** Where are the inverted faults?
- **Answer:** The inverted faults can be seen in the DIC minimum normal strain figures. It would be too uncertain to draw them at the topographic profiles without having access to cross-sections. We also believe the image would become too crowded when adding faults to them, and we already pointed out the new reverse faults.

**Line 295**

- **Comment:** Figure caption
- **Answer:** Thanks for the suggestion, it is modified.

**3.2.2 Oblique rifting - orthogonal (B3) and oblique inversion (B4)**

**Line 302**

- **Comment:** oblique (B4)
- **Answer:** Thanks for the suggestion, it is modified.

**Line 304**

- **Comment:** échelon to be corrected everywhere in the text
- **Answer:** Thanks for the suggestion, it is modified.

**Line 305**

- **Comment:** show
- **Answer:** Thanks for the suggestion, it is modified.

**Line 308**

- **Comment:** formation of a
- **Answer:** Thanks for the suggestion, it is modified.

**Line 312**

- **Comment:** was
- **Answer:** Thanks for the suggestion, it is modified.

**Line 313**

- **Comment:** Quantification of all this description as for models B1 and B2
- **Answer:** Thanks for the suggestion, we were more specific with the timing of the models through the description.

**Line 315**

- **Comment:** Model
- **Answer:** Thanks for the suggestion, it is modified.

**Line 317**

- **Comment:** Shows
- **Answer:** Thanks for the suggestion, it is modified.

**Line 322**

*"The topography profiles indicate uplift of the rift structures (17 mm elevation of the bottom of the graben) and the new reverse faults on both sides of it (Fig 5p), and while the northern reverse fault became inactive, distributed uplift affected the northern part of the model (Fig. 5p)."*

- **Comment:** It cannot be northern in both cases
- **Answer:** Both are northern because we were describing the distributed uplift related to the reverse fault inactivity (even though the fault is inactive, there is still some general uplift going on).

**Line 322**

- **Comment:** Topographic
- **Answer:** Thanks for the suggestion, it is modified.

**Line 324 (figure 5)**

- **Comment:** Where are the inverted normal faults?
- **Answer:** The reactivation of normal faults can be seen on the DIC minimum normal strain analysis figures related to the experiments (Figure 5l)

**Line 325**

- **Comment:** Figure caption
- **Answer:** Thanks for the suggestion, it is modified.

**Line 330**

- **Comment:** This makes it impossible to compare final stages. Why did you do this?
- **Answer:** See previous comments on this topic. This is not a major problem to our models and to the kind of assessment we are doing in this manuscript.

**3.3.1 Orthogonal rifting with sedimentation - orthogonal (C1) and oblique inversion (C2)**

**Line 338**

- **Comment:** oblique (C2)
- **Answer:** Thanks for the suggestion, it is modified.

**Line 345**

*"Cross-section thickness measurements from each of the 15 minutes syn-rift sedimentation intervals (I1-I8), indicate a progressive increase of subsidence in the first two sedimentation intervals (Fig. 7a$_I$; I1 to I3)."*

- **Comment:** How did you measure?
- **Answer:** It was measured at the cross-sections sedimentation intervals represented by the quartz and feldspar sand intercalation, indicated in figure 7. We have modified the text and the figure to be more specific.

- **Comment:** Indicate where the reader can see this (insets in Fig. 7)
- **Answer:** Thanks for the suggestion, it is modified.

**Line 349**

- **Comment:** Inherited
- **Answer:** Thanks for the suggestion, it is modified.

**Line 351**

- **Comment:** also rose
- **Answer:** Thanks for the suggestion, it is modified.

**Line 356**

- **Comment:** SE quadrants
- **Answer:** Thanks for the suggestion, it is modified.

**Line 357**

- **Comment:** Shows
- **Answer:** Thanks for the suggestion, it is modified.

**Line 358**

- **Comment:** visible on
- **Answer:** Thanks for the suggestion, it is modified.

**Line 361 (figure 6)**

- **Comment:** Minimum Correct everywhere
- **Answer:** Thanks for the suggestion, it is corrected.

**Line 363**

- **Comment:** Figure caption
- **Answer:** Thanks for the suggestion, it is modified.

**Line 369 (figure 7)**

- **Comment:** Replace with: Model C1 - orthogonal rift and inversion. Idem for b, c and d
- **Answer:** Thanks for the suggestion, it is modified.

- **Comment:** Move inset upwards so that we can see the full reverse fault in the N
- **Answer:** Thanks for the suggestion, it is modified.

**Line 370**

- **Comment:** Figure caption
- **Answer:** Thanks for the suggestion, it is modified.

**3.3.2 Oblique rifting with sedimentation – orthogonal (C3) and oblique (C4) inversion**

**Line 385**

- **Comment**: results similar to models
- **Answer:** Thanks for the suggestion, it is modified.

**Line 386**

- **Comment:** rifting,
- **Answer:** Thanks for the suggestion, it is modified.

**Line 390**

- **Comment:** Represent, filling
- **Answer:** Thanks for the suggestion, it is modified.

**Line 391**

- **Comment:** be more specific
- **Answer:** Thanks for the suggestion, it is modified.

**Line 406**

- **Comment**: hidden by the inset
- **Answer:** Thanks for the observation, it is corrected.

- **Comment:** Not necessarily. It can root at the inherited normal fault immediately to the South
- **Answer:** We wrote what was observed on the cross-section and there was no connection with the normal fault. However, we improved the description in the text, thank you for the observation

**Line 408**

- **Comment:** Figure caption
- **Answer:** Thanks for the suggestion, it is modified.

**4 Discussion**

**4.1 Summary and comparison to previous models**

**Line 417**

- **Comment:** imposed kinematics
- **Answer:** Thanks for the suggestion, it is modified.

**4.1.1. Rifting phase**

**Line 427**

*"Furthermore, oblique extension caused a decrease in graben width compared to the orthogonal rifting models, as also described in previous studies (Zwaan and Schreurs, 2016; Zwaan et al., 2018a) (Figs. 3 and 4)."*

- **Comment:** The issue is not that it is oblique, it is that run time is the same for orthogonal and oblique rifting
- **Answer:** We are not sure what is the issue here. We have the same amount of displacement along the direction of divergence used in the rifting phase of each model.

**Line 429**

*"This reduction in width is caused by the strike-slip component accommodating deformation in oblique rifting settings."*

- **Comment:** It seems to me that it is due to different amounts of extension. To be comparable, the orthogonal extension should be identical in both models, which means that the oblique extension should run for longer time.

- **Answer:** See previous comments, our analysis is consistent.

**Line 431**

- **Comment**: In science, demonstrations are restricted to Mathematics. Replace with showed

- **Answer:** Thanks for the suggestion, it is modified.

**Line 434**

*"A narrower graben forming during oblique rift evolution led to smaller loads of sedimentation, consequently there was less weight to cause graben floor subsidence."*

- **Comment:** But not because it is oblique. Again the problem of identical time for differently otiented vectors

- **Answer:** Again, it is not a problem. Same answer as to the same questioning above (lines 427, 429).

**4.1.2. Inversion phase**

**Line 460**

*"Without sedimentation, the rift structures were reactivated during inversion, and the new reverse faults developed independently of inversion direction (Fig. 9)."*

- **Comment:** This is not true. Compare panels h and p in Fig. 4, and h and p in Fig. 5. If you draw the inverted normal faults you will see significant differences

- **Answer:** In figure 9 (schematic for models in figures 4 and 5) we show those topographic profiles at the 3D cubes for each model and we see new reverse faults developing in every model. As for the inverted normal faults, we see the reactivation in the DIC and its very clear how the topography inside the graben is inverted.

- **Comment**: These are sketches, not true profiles as in Figs. 4 and 5
- **Answer:** Yes, they are the 3D sketches of the true topographic profiles in figures 4 and 5.

**Line 464**

- **Comment:** You must show this on the topographic profiles by drawing the inverted faults
- **Answer:** Reactivation is shown in the DIC figures of minimum normal strain and since we don't have cross-sections, we don't want to draw these faults in section. That's why we added dashed lines in the schematic drawings.

**Line 475**

- **Comment:** This is one reason why Marques et al. (2014) considered a much smaller angle (in Brun and Nalpas, 1996, notation) for the inversion of the Araripe Basin. Fault lubrication can also promote inversion of normal faults, as experimentally shown by Marques and Nogueira (2008), and observed in the field by Marques et al. (2014)
- **Answer:** See earlier replies: the argument of having more oblique convergence or fault lubrication/weakening is not relevant, as there are no large-scale inverted rift faults in the area.

**Line 476**

- **Comment:** If you believe that these authors are corrected, why did you use 45º?

- **Answer:** We were interested in the general effect of obliquity to explore the potential evolution of the basin, and what is pointed out here is that similar behaviour has been observed by other modellers. This is a good thing in our eyes.

**Line 479**

- **Comment:** In Fig. 4p I can clearly see inverted normal faults in the graben. Why don't you show them here?
- **Answer:** We are showing them, because these schematic drawings of the models are exactly the topographic profiles you see in the results figures. The top lines in the front and in the back of the cube are the topographic lines for each model. But we now added some detail to the figure.

- **Comment:** The same here. Check Fig. 5p.
- **Answer:** The problem we see here is that we did not add the dashed lines representing the probable reactivated normal faults we see in the DIC. We have now added this to the figure so it's the same as models B1, B2 and B3.

**4.2 Comparing model results with the Araripe Basin**

**Line 487**

- **Comment:** currently peaks
- **Answer:** Thanks for the suggestion, it is modified.

**Line 488**

- **Comment:** 2007), i.e. ca. 500 m above the surrounding basement.
- **Answer:** Thanks for the suggestion, it is modified.

**Line 489**

- **Comment:** and differential erosion
- **Answer:** Thanks for the suggestion, it is modified.

**Line 499**

*"However, although we observed some fault reactivation in our oblique inversion models, this reactivation did never lead to full inversion of the graben normal faults (Figs. 9 and 10), which contradicts the Marques et al. (2014) scenario."*

- **Comment:** No, it does not. We proposed a much lower angle between sigma 1 and graben strike. Besides, we also considered fault weakening as a facilitator mechanism, as experimentally shown by Marques and Nogueira (2008)
- **Answer:** Deleted part of the sentence the reviewer does not agree with, but as pointed out before, no indications of large-scale inversion of rift faults is observed in other studies of the basin.

**Line 499**

*"In fact, no large-scale fault reactivation has been observed in the Araripe Basin (Ponte and Ponte-filho, 1996)."*

- **Comment:** Maybe Ponte and Ponte-Filho (1996) overlooked large-scale fault reactivation, but we did not, and we show pictures of them in our paper Marques et al. (2014)
- **Answer:** To be frank, inversion on the scale proposed by Marques et al. (2014) would be somewhat hard to miss on seismic data. Instead, it may be more likely that the faults found by Marques et al. (2014) are not that significant as the authors propose. See also previous replies on this topic

**Line 500**

*"A further argument against the Marques et al. (2014) scenario would be that the post-rift sediments outside the original graben domain would not have been uplifted in contrast to what we see in nature (Fig. 1)."*

- **Comment:** This is a wrong and unfair statement. For the study in 2014, we did not have the time to study the rocks outside the main graben. In the Rio do Peixe Basin we had plenty of time and so we found reverse faults outside the main basin. You can check Vasconcelos et al. (2020), for instance in Fig. 9a and e.

- **Answer:** The reviewer is right and are thankful for the reference to this paper. Vasconcelos et al (2021) found reverse faults in the basement outside the Rio do Peixe Basin, which fits perfectly with our models. We are adding this in the discussion. On the other hand, in the inversion model by Marques et al. (2014), shown in their figure 18, all deformation is concentrated in the form of inverted normal faults (which, as pointed out earlier, does not fit with data from the Araripe Basin).

- We think we were not clear with this sentence, so we deleted it and wrote a new paragraph. What we meant here was that part of the Araripe high-standing topography (post-rift units) is not only on top of the previous rift grabens of the basin, but in the western part of the Araripe mesa the post-rift units are covering the pre-Cambrian basement. Therefore, we should expect to see a structural difference along this topographic feature if this were to be a result of pure inversion of the original rift faults (it would mean that the post-rift sediments away from the original graben would not have been uplifted).

- About Rio do Peixe Basin, there are no post-rift units currently there (maybe they are simply eroded), which is probably why there is no high-standing topography there too. Vasconcelos et al (2021) state that the inversion intensity in the Rio do Peixe faults is mild to moderate compared to the Araripe Basin (even so, the authors found reverse faults away from the basin, which makes their presence in the more inverted Araripe Basin all the more likely). However, the main difference between these two basins seems to be the current presence of the post-rift units, and without these units Araripe probably would look the same as the Rio do Peixe Basin topography wise.

**Line 503**

*"This uplift of post-rift sediments outside of the original graben domain can be explained by the Peulvast and Bétard (2015) scenario…"*

- **Comment:** Not only. It can also be explained by thrusting outside the main basin, as proposed by Vasconcelos et al. (2021) for the Rio do Peixe Basin

- **Answer:** Indeed, this is the whole point: we expect reverse faults outside of the basin, as seen in our models. We thank the reviewer for pointing us to the Vasconcelos et al. (2021) paper that supports our interpretation (see answers to the previous comment[s]). We rewrote the text a bit here.

**Line 508**

- **Comment:** Many other references are missing here. See list given in the report

- **Answer:** Added the references listed about the South American plate compression.

**Line 510**

- **Comment:** Critical references are missing here, including for the Araripe Basin inversion. See list given in the report

- **Answer:** Added the references related to the basin inversion proposed by the reviewer.

**Line 514**

- **Comment:** This is simply wrong, for two reasons: (1) you did not test Peulvast and Bétard's hypothesis; (2) you did not test what Marques et al. (2014) proposed for the Araripe inversion, which is low inversion angle and fault lubrication.

- **Answer:** We deleted this sentence and rewrote it focusing on a modified argument (see previous replies).

**Line 518**

- **Comment:** This statement is wrong, because you only tested 45º

- **Answer**: Rewrote the sentence to better explain our point here. We were only talking about our own models and not generalizing.

**Line 519**

- **Comment:** This is a wrong statement, because, to my knowledge, reverse faults have not been observed outside the main Araripe Basin
- **Answer:** As the author specified in another comment, Marques et al. (2014) did not have the opportunity to explore the geology away from the basin. This does not mean that such structures do not exist. In fact, as the reviewer pointed out, such faults have been found in the nearby Rio do Peixe Basin and our modelling results suggest that they may very well exist in the Araripe Basin area as well.

**5 Conclusions**

**Line 538**

- **Comment:** This is a conclusion already reached by several authors before you, so it is not a conclusion of your work
- **Answer:** It is an important outcome of this study and should be mentioned here.

**Line 542**

- **Comment:** I do not understand this. I cannot see what you mean in terms of mechanics
- **Answer:** What we mean is that when comparing an empty graben with a filled one (like in nature) we don't see much fault reactivation after compression.

**Line 546**

- **Comment:** Again, this is simply wrong. Read what Marques et al. (2014) said about the inversion of the Araripe Basin
- **Answer:** See previous comments explaining why previous models do not fully explain the situation in the Araripe Basin.

---

## Author Comment (AC2)

**Referee 2 - Ioan Munteanu**

**Comment:** Basin inversion pccurs aslo as extensional not just as compresional. Please mention that you refere to compresional inversion of extensional basins.

**Answer:** Thank you for your comment. In this manuscript, we talk about basin inversion, which means a phase of (oblique) extension, followed by a phase of (oblique) shortening. We believe the reviewer is referring to "negative inversion", which means that you first have contractional structures such as reverse faults and thrusts that have been reactivated during subsequent extension, whereas "positive inversion" means one has first extensional structures that are subsequently affected by shortening. However, here we talk about "basin inversion", and we believe this can only mean that you have first a basin, and then reactivation of the basin due to (oblique) shortening. However, in order to avoid any confusion, we now specify that the basin inversion in the manuscript refers to positive inversion.

**Line 11**

- **Comment:** Basin inversion occurs either in extensional or compressional settings, like negative or positive inversion.
- **Answer:** This comment is not very clear to us

**Line 42**

- **Comment:** This aborted rifts are actually part of the early intra-continental stage. And like North sea is actually part of the Atlantic system.
- **Answer:** Thank you for your comment, this is more or less what we have written in the text

**Line 47**

- **Comment:** I can't see this E-W direction in your fig. 1
- **Answer:** Thanks for the suggestion, it is modified.

**Line 56**

- **Comment:** A rift can't push so much that you invert a a basin. The formation of oceanic crust will bring exhumation especially on the rift shoulder. Other must be the case

- **Answer:** It is not only because of that, it's the combination of the mid Atlantic ridge push and Andes mountains initial subduction to the west

**Line 77 – figure 1**

- **Comment:** Can you have an seismic like or a geological cross-section to illustrate the inversion?

- **Answer:** We can't reproduce the seismic lines in this manuscript due to copyright restrictions, but we cited a recent work in the discussion that show two interpreted seismic lines for the Araripe Basin.

- **Comment:** The offset of this fault is similar with the one in the extensional stage, where is the inversion?

- **Answer:** This is a representation of the inversion model proposed by Marques et al. (2014), where, according to the authors, previous rift faults went under large-scale inversion. The offset is clearly different from the rift stage though, and the proposed reactivation of rift faults is indicated by arrows.

**Line 236**

- **Comment:** and how much extension?

- Answer: this is after 30 minutes, so that is 10 mm of divergence, given a divergence velocity of 20 mm/h. We have added some quantification of divergence and convergence wherever we felt it would be good to do so.

**Line 243**

- **Comment:** Is better to represent this in extension rate etc.

- **Answer:** We agree that it may be better to specify the amount of divergence for a given time step and have added these details (see also previous comment)

**Line 247**

- **Comment:** Which means in % of extension
- **Answer:** What is specified is the width of the graben. The total divergence at that time step is 40 mm (see also previous comments)

**Line 280**

- **Comment:** As I stated earlier, will be easy to quantify also in % relative to you crust
- **Answer:** The original sand layer (upper crust) is 6 cm thick, so the subsidence is similar to 33% of the thickness of the upper crust. Note however, that the lower crust is also rising up below the graben (see Fig. 3), so that the total thickness of the sand layer is ca. 33% of the original at this point. We have added some quantification here and elsewhere where we thought it helpful.

**Line 304**

- **Comment:** you want to say relay ramps. En enchelon we use more for strike-slip, which is not the case.
- **Answer:** En echelon is also routinely used for oblique extension settings, we prefer to keep it as is.

**Line 502**

- **Comment:** This natural case scenario has to be supported by an geological cross-section, the sketch in the Figure 1 is not enough
- **Answer:** We don't have field data, this is not a field study, and we can't reproduce seismic sections. We are proposing a model based on the experiments and it fits with the general data on the area. We believe this is ok.

**Line 513**

- **Comment:** reactivation or inversion?
- **Answer:** Thanks for the suggestion, it is modified with inversion.

**Line 514**

- **Comment:** Inversion
- **Answer:** Thanks for the suggestion, it is modified.

---

## Referee Report (RR1)

**Report on** "Analogue modelling of basin inversion: the role of oblique kinematics and implications for the Araripe Basin (Brazil)" by Richetti et al.

**Assessment**: the topic of the ms. is relevant for geosciences, and therefore suitable for EGUsphere, but not in its present form. The main problems to be addressed are: (1) lack of quantification of length (so far only qualitative); (2) novelty (what is actually new? Think very carefully); (3) misleading use of the experimental results (do not say that there are no signs of inversion, as in the Abstract, when we can see them clearly in the submitted figures); (4) comparison between experiments and nature (find a length ratio); (5) comparison with previous work (you cannot compare onions with potatoes, and you should read previous literature more carefully, e.g. Marques et al., 2014, and Rosa et al., 2023). If these problems are not properly solved, I do no think this ms. should be accepted for publication because it comprises fatal flaws.

**Main comments**

1. The title of the ms. is misleading in two ways: (1) the first part of the title "*Analogue modelling of basin inversion: the role of oblique kinematics*", because obliquity was not fully tested, i.e. from 0 to 90º, only the 90 and 45º that have long been shown unfavourable to invert high-angle, high-friction precursory normal faults; and (2) the second part "*implications for the Araripe Basin (Brazil)*" because the authors did not test the relevant variables and parameters, and, therefore, cannot directly compare the experiments with the AB.

2. To explore how tectonic basin inversion in the AB could have taken place, you would need to: (1) reproduce the natural example in the rift phase, which you did not accomplish entirely,

because you only produced one set of parallel faults, and the AB has at least three main sets (E-W, NE-SW, and NW-SE (e.g. Rosa et al., 2023). (2) Use the full range of shortening directions, because it is long known that high-angle convergence (> 45°) does not produce inversion in high-angle precursory normal faults. (3) Test fault rock rheological properties, e.g. viscous behaviour materials as observed in the AB (clays and evaporites). Unfortunately, you did not test any of these variables and parameters. What you tested (shortening angle) has been tested and theoretically explained long ago. Not having tested the most relevant variables and parameters that can be responsible for the inversion of high-angle precursory normal faults, you have no argument to claim that Marques et al.'s hypothesis is wrong.

3. Inversion of graben faults must be quantified in mm, and faults location must be shown on the topography graphs. Figure captions must include the amount of vertical exaggeration. How do you explain inversion with shortening at 90° that we can see in all topography graphs and model sections (Fig. 7)? This is critical to your work.

4. How can you compare amounts of inversion in model and nature if you do not define the length ratio? How many meters in nature for each millimetre in the model? If we take the value of 1,600 m for the depth of the AB (de Castro and Branco, 1999) and the ca. 20 mm depth in the models, then we have a length ratio of 1.25E-5. This means that 1 mm in the model corresponds to 80 m in nature (scale = 1/80,000). If we use this ratio in Fig. 6h, for example, we can see that the model topography is greatly exaggerated compared to nature, because it is more than 3 times (ca. 1730 m) the 500 m in the AB. For the graben to vanish between the rift and inversion stages, and stand out of the topography at the end stage, the graben must be uplifted by ca. 6.8 mm, i.e. ca. 550 m in nature, which corresponds to the actual altitude of the AB relative to the host basement altitude (ca. 500 m). This is the opposite of your conclusion that tectonic inversion

cannot explain the AB. Now the problem is to explain inversion with high angle shortening (including 90º), which could comprise the novelty of your work. The authors should also say that the inversion structures found by Marques et al. (2014) in the host basement outside the basin were also found in the experiments, but disproportionate in height to what we observe in nature. If you read Marques et al. (2014) carefully, you will see that they propose reverse faulting outside the current Araripe Basin. You can confirm that in section 3.3.2.3 and Figs. 15 and S1. However, you never mention this in your text, especially when discussing the experimental reverse faulting outside the basin and relation to what is known in the AB.

5. What are the effects of deformation of the foam/plexiglass base on the observed strain in sand? This seems to me critical to the partial understanding of the experimental results.

Lines 47-48 – "*The rift structures of the Araripe Basin mainly strike NE-SW*": This is not true, because the main boundary fault is E-W, by brittle reactivation of the Precambrian Patos Shear Zone. In your experiments, the NE-SW structures are not even faults, they are "strain bands" as you call them (e.g. Fig. 5).

Lines 66-67 – "*The Peulvast and Bétard (2015) scenario fits with the general absence of large-scale inversion of normal faults as seen on seismic sections from the Araripe Basin (Ponte and Ponte-filho, 1996, Rosa et al., 2023*": The 1996 reference is missing in the references list, and I could not even have access to it through my Brazilian colleagues. If the reader cannot have access to these seismic data, you cannot use them as argument. Regarding Rosa et al. (2023), they only show two and very short seismic lines. Interestingly, you can see good signs of inversion in one of the lines. In fact, Rosa et al. (2023) report important signs of tectonic inversion in the Araripe Basin. They simply interpret them differently from Marques et al. (2014).

Line 77 – "… *novel set-up*": Where is the novelty?

Line 119 – "*model set-up … fundamentally different*": Why is that so? What changes? What are the effects on final results?

Lines 141-142 – "*… 6 cm thick layer of fine quartz sand … representing a 20 km brittle upper crust*": If 60 mm in the model correspond to 20E6 mm in nature, then L* = 3E-6. This means that 1 mm in the model equals 333 m in nature. Given that the average graben in your experiments is ca. 20 mm deep, this scales up to nature to 20x333 = 6660 m, which is more than 4 times the 1600 m proposed by de Castro and Branco (1999)

Lines 307-308 – "*… localized strain both along the intra-graben faults …*": How do you explain intra-graben inversion by orthogonal shortening? This is critical to your work.

Figs. 4 and 5 – several features can be measured on the topography graphs, which deserve explanation.

Fig. 6 – panels g and h show that the graben has vanished from the rift to the inversion phases; how do you explain this? Besides, there is good evidence in panel f for inversion of the N master rift fault (sharp step in blue shades).

Fig. 7 – in panels a and b you must give the references of the syn-rift layers on both sides of the faults so that we can evaluate the amount of inversion.

Lines 377-378 – "*… while no reactivation is visible in the inherited rift structures*": Then how do you explain that the initial graben (panel g) has vanished (panel h) in Fig. 6? The same applies to Fig. 8.

Many comments, main and minor, can be found in the attached annotated PDF.

The text still needs revision of the English.

Lisbon, 28.04.2023

Fernando Ornelas Marques

[revised manuscript text omitted]

---

## Author Response (AR2)

Dear authors,

Thank you for submitting "Analogue modelling of basin inversion: the role of oblique kinematics and implications for the Araripe Basin (Brazil) to Solid Earth. I received a detailed evaluation of the revised version of your manuscript. As you see R1 still raises substantial concerns related to the novelty of the paper, quantitative aspects of your study, consistency between text and features shown in the figures and the interpretation of the results in context of explaining the Araripe Basin inversion.

- **Answer:**
  - We thank you and reviewer 1 for considering our manuscript. Please find detailed replies below

Though R2 exposes some weaknesses, which need to be addressed to make the manuscript suitable for publication in Solid Earth, the review also provides clear guidance on the critical issues to focus on and hence provides the pathway to improvement.

- **Answer:**
  - We did not receive any files from a second reviewer. We contacted Solid Earth, but they informed us no second review was received. As such, we focus on the comments from reviewer 1.

It would probably be most efficient to start with comments 3 & 4 under "Main comments" as these form the foundation for the other comments. Consistency between modelling results described in the text and the structural and topographic elements in the figures needs to be warranted.

- **Answer:**
  - We believe that various of the issues raised by the reviewer seem to stem from a misunderstanding of the text and our model series. We have provided detailed answers as to why we believe that our manuscript needs only limited modification.

Based on the above, I am returning the manuscript so that you can make the necessary changes.

Looking forward to receiving the modified version of your manuscript.

Ernst Willingshofer

**Referee 1 -** Fernando Ornelas Marques

*Assessment: the topic of the ms. is relevant for geosciences, and therefore suitable for EGUsphere, but not in its present form. The main problems to be addressed are: (1) lack of quantification of length (so far only qualitative); (2) novelty (what is actually new? Think very carefully); (3) misleading use of the experimental results (do not say that there are no signs of inversion, as in the Abstract, when we can see them clearly in the submitted figures); (4) comparison between experiments and nature (find a length ratio); (5) comparison with previous work (you cannot compare onions with potatoes, and you should read previous literature more carefully, e.g. Marques et al., 2014, and Rosa et al., 2023). If these problems are not properly solved, I do no think this ms. should be accepted for publication because it comprises fatal flaws.*

**Main comments**

1. The title of the ms. is misleading in two ways: (1) the first part of the title "Analogue modelling of basin inversion: the role of oblique kinematics", because obliquity was not fully tested, i.e. from 0 to 90°, only the 90 and 45° that have long been shown unfavourable to invert high-angle, high-friction precursory normal faults; and (2) the second part "implications for the Araripe Basin (Brazil)" because the authors did not test the relevant variables and parameters, and, therefore, cannot directly compare the experiments with the AB.

- **Answer:**
  - **Point 1:** we disagree as we do most certainly test oblique extension, vs. orthogonal extension, and the modelling approach is reasonable as explained in our answer to point 2

o **Point 2:** we must point out that all (analogue) models are simplifications of natural systems, used to test specific parameters. In our models we run 45˚ oblique rifting models vs. orthogonal extension models, which is inspired by hypotheses regarding plate motion direction during Araripe Basin formation and later basin inversion. As we explain in the manuscript, the general insights derived from our models fit quite well with observations from nature, so we believe that this objection by the reviewer does not hold.

2. To explore how tectonic basin inversion in the AB could have taken place, you would need to: (1) reproduce the natural example in the rift phase, which you did not accomplish entirely, because you only produced one set of parallel faults, and the AB has at least three main sets (E-W, NE-SW, and NW-SE (e.g. Rosa et al., 2023). (2) Use the full range of shortening directions, because it is long known that high-angle convergence (> 45º) does not produce inversion in high-angle precursory normal faults. (3) Test fault rock rheological properties, e.g. viscous behaviour materials as observed in the AB (clays and evaporites). Unfortunately, you did not test any of these variables and parameters. What you tested (shortening angle) has been tested and theoretically explained long ago. Not having tested the most relevant variables and parameters that can be responsible for the inversion of high-angle precursory normal faults, you have no argument to claim that Marques et al.'s hypothesis is wrong.

- **Answer:**
  - o **Point 1:** again, models are a simplification of reality. Trying to simply reproduce all aspects of the natural example may look nice, but hampers a proper understanding of the actual processes since it will become near impossible to understand the impact of a given parameter. As such, it is much better to design simpler models. The models we completed show first-order insights that we can apply to better understand the general inversion history of the Araripe Basin.
  - o **Point 2:** We did test a number of parameters and believe that our model results are sufficiently robust to support our interpretation of the Araripe Basin. Running additional models is therefore beyond the scope of our study. It is in fact reassuring that our results, obtained with a new set-up, fit with previous modelling results that involved other set-ups, we do not

see how this can be a weakness here (we see it as a confirmation that our results are valid).

- o **Point 3:** Again, we test large-scale processes. It is true that similar insights have been provided by previous modellers, but we apply a different model set-up than was previously used. Furthermore, the fact that our model results fit well with previous model results supports their robustness, rather than proving its weakness.

  - ▪ Regarding the inversion of high-angle faults: as we point out in our manuscript, there is precious little evidence for large-scale inversion of high-angle faults in the Araripe Basin. As such, the whole discussion regarding inversion of rift-related normal faults is rather irrelevant. Our models show that instead of inverting the original rift faults, both orthogonal and oblique inversion should result in the development of new reverse faults instead. As we point out in our manuscript, there is good evidence in the region to support this new interpretation, over the previous Marques et al. 2014 interpretation that invokes large-scale inversion of initial high-angle normal faults.

    - • NB: Marques et al. (2014) do mention the observation of some reverse faults outside the Araripe Basin, but in their discussion they clearly promote the concept of large-scale inversion of normal faults. We have added this to the discussion.

3. Inversion of graben faults must be quantified in mm, and faults location must be shown on the topography graphs. Figure captions must include the amount of vertical exaggeration. How do you explain inversion with shortening at 90º that we can see in all topography graphs and model sections (Fig. 7)? This is critical to your work.

- • **Answer:**
  - o We are sorry to answer a bit harshly here, but we believe that the reviewer is nitpicking here, the structures are perfectly well visible in the images we prepared. Also, a vertical scale is already provided in the figures, next to the horizontal scale, but we have added a mention of the vertical exaggeration to the captions. It is not clear what the reviewer means with

the last part of the comment. Inversion in all models leads to the development of reverse faults and uplift of the original rift basin + surrounding material along these reverse faults.

- NB: there seems to be some topography along the axis of the model in the models involving orthogonal inversion, which seems to suggest minor inversion of previous rift normal faults. This is an artifact of manually filling in the basin during rifting, leaving a higher topography due to minor filling errors (the model was not scraped flat after each sedimentation interval). DIC results show no indication of reactivation of these faults, and even if there would indeed be some inversion of normal faults, it is really not much compared to the overall displacement in the system and thus not significant (a boundary effect).

4. How can you compare amounts of inversion in model and nature if you do not define the length ratio? How many meters in nature for each millimetre in the model? If we take the value of 1,600 m for the depth of the AB (de Castro and Branco, 1999) and the ca. 20 mm depth in the models, then we have a length ratio of 1.25E-5. This means that 1 mm in the model corresponds to 80 m in nature (scale = 1/80,000). If we use this ratio in Fig. 6h, for example, we can see that the model topography is greatly exaggerated compared to nature, because it is more than 3 times (ca. 1730 m) the 500 m in the AB. For the graben to vanish between the rift and inversion stages, and stand out of the topography at the end stage, the graben must be uplifted by ca. 6.8 mm, i.e. ca. 550 m in nature, which corresponds to the actual altitude of the AB relative to the host basement altitude (ca. 500 m). This is the opposite of your conclusion that tectonic inversion cannot explain the AB. Now the problem is to explain inversion with high angle shortening (including 90º), which could comprise the novelty of your work. The authors should also say that the inversion structures found by Marques et al. (2014) in the host basement outside the basin were also found in the experiments, but disproportionate in height to what we observe in nature. If you read Marques et al. (2014) carefully, you will see that they propose reverse faulting outside the current Araripe Basin. You can confirm that in section 3.3.2.3 and Figs. 15 and S1. However, you never mention this in your text, especially when discussing the experimental reverse faulting outside the basin and relation to what is known in the AB.

- **Answer:**
  - The reviewer seems to demand and exact reproduction of the Araripe Basin in the sandbox, which he, as an experienced analogue modeler, must know is impossible. As explained before, we present simple models to try and understand the impact of oblique kinematics on the evolution (and inversion) of the Araripe Basin. We provide a first-order interpretation that fits well with observations from the area, which we believe should be quite clear from our manuscript.
  - Scaling is clearly provided in the text (3 cm = 10 km).
  - Regarding the 6.8 mm inversion the reviewer seems to identify in our models: the reviewer seems to have misunderstood the models. After rifting, the basin is filled so that the model topography is (more or less) flat again. During inversion, the basins are inverted by the development of new reverse faults, which cause the uplift of the basin and its surrounding "basement". In no case do we see the 6.8 mm inversion accommodated by normal fault reactivation the reviewer argues for.
  - We have re-read the Marques et al. 2014 paper in detail. In their results section, the authors indeed mention some faults in the basement. We added a reference to these faults in our discussion, which does nicely support our interpretation of how inversion in the Araripe Basin could have been achieved by the development of new large-scale and low-angle reverse faults.
    - However, we must point out that in their discussion, Marques et al. (2014) explicitly champions the (dominant) role of the inversion of normal faults (while ignoring the potential [dominant] contribution of the new reverse faults the did observe in the field). The large-scale inversion of rift faults as proposed by Marques et al. (2014) is simply not visible on seismic data, as discuss in our manuscript.

5. What are the effects of deformation of the foam/plexiglass base on the observed strain in sand? This seems to me critical to the partial understanding of the experimental results.

- **Answer:**

o The foam/plexiglass base creates a distributed stretching boundary condition at the base of the model, which can be oriented either in an orthogonal or an oblique direction to induce either orthogonal or oblique rifting, respectively. Vice versa, orthogonal and oblique inversion can be induced. This deformation is transmitted through the viscous layer into the overlying sand layer. In this sand layer, we apply a weakness (a seed) to localize a rift basin along the central axis of the model. This set-up is very different from set-ups using base plates or moving basement blocks, as it allows the system more freedom to develop. Also, the viscous layer acts as a buffer layer that evenly distributes the velocity field and dampens potential displacement/strain heterogeneities from the basal setup. We tested this setup using DIC once, and it clearly showed that (for those velocities we are using) the surface of the viscous layer deforms homogeneously. Even so, we get similar results to previous modelling studies, which indicates the validity of our approach.

▪ For more details on the various set-ups used for analogue modelling, and the uniqueness of our model set-up, see the review paper by Zwaan et al. 2022 in Solid Earth

• Link: https://doi.org/10.5194/se-13-1859-2022

Lines 47-48 – "The rift structures of the Araripe Basin mainly strike NE-SW": This is not true, because the main boundary fault is E-W, by brittle reactivation of the Precambrian Patos Shear Zone. In your experiments, the NE-SW structures are not even faults, they are "strain bands" as you call them (e.g. Fig. 5).

• **Answer:** Araripe Basin is a system of half-grabens mainly controlled by normal faults striking NE-SW. The Patos Shear Zone is the main E-W shear zone limiting (more or less) the present-day north limit of the basin; however, the grabens within the Araripe Basin are mainly controlled by the other NE-SW Precambrian shear zones connecting the E-W Pernambuco in the south and Patos shear zones in the north. We have modified the sentence to avoid any further confusion.

o As we clearly write in the manuscript, the models develop in series of en echelon faults that represent the basin orientation within the Araripe Basin structure. The "strain bands" are later structures within the basin, that also

follow the general orientation of intra-basin structure. We do not really follow how that is an issue. As said before, we are interested in the large-scale development of inversion in the Araripe Basin, and these model features are quite compatible with both oblique rift kinematics as with the structural orientations in the Araripe Basin.

Lines 66-67 – "The Peulvast and Bétard (2015) scenario fits with the general absence of large-scale inversion of normal faults as seen on seismic sections from the Araripe Basin (Ponte and Ponte-filho, 1996, Rosa et al., 2023)": The 1996 reference is missing in the references list, and I could not even have access to it through my Brazilian colleagues. If the reader cannot have access to these seismic data, you cannot use them as argument. Regarding Rosa et al. (2023), they only show two and very short seismic lines. Interestingly, you can see good signs of inversion in one of the lines. In fact, Rosa et al. (2023) report important signs of tectonic inversion in the Araripe Basin. They simply interpret them differently from Marques et al. (2014).

- **Answer:**
  - We have now added the Ponte and Ponte-filho (1996) reference, which is a regularly cited work in the context of the Araripe Basin (it is for instance also cited in Marques et al. 2014). As such, we do not see why we cannot refer to this publication in our manuscript. Link: https://www.researchgate.net/publication/355575301_Estrutura_Geologica_e_Evolucao_Tectonica_da_Bacia_do_Araripe
  - The inversion of initial rift faults reported by Rosa et al. (2023) is not "simply" a different interpretation; as we discuss in the manuscript, they present a fundamentally different interpretation: that these inverted normal faults are related to a shift in kinematics during the rifting phase, and not to later inversion. As showed by the very representative seismic lines of the Araripe basin, this inversion of rift faults is only present in the rift formations and does not affect the post-rift formations. If there was a high degree of normal fault inversion of Araripe Basin as proposed by Marques et al (2014), it should be very clearly seeing in any of the seismic lines, which it is not.

o Instead, we present a different interpretation of inversion in the Araripe Basin involving the formation of new reverse faults to explain the present-day situation. We believe that our interpretation does fit the observations obtained from both our model and nature.

Line 77 – "… novel set-up": Where is the novelty?

- **Answer:**
    o The novelty is that the set-up is the use of a foam and Plexiglass base, which was until now only used for (oblique) rifting experiments, and now for the first time for inversion experiments (see also the Zwaan et al. 2022 review paper). In contrast to previous inversion modelling works, we use a foam-plexiglass base that induces distributed deformation at the base of the model. As such, deformation the model is less directly constrained than in more traditional base plate model (see also previous reply).
        ▪ Note that the details of the set-up are addressed later in the methods, and the reader should not expect to find all details here in these lines in the introduction.

Line 79 - I do not understand this rationale, because these angles are long known to work against tectonic inversion in high-angle normal faults

- **Answer:**
    o We apply a new set-up for basin inversion modelling that also show that orthogonal inversion counteracts normal fault reactivation. This is a good result we thing (see also previous replies).
    o The kinematics applied in our modelling study were inspired by the kinematics proposed for the Araripe Basin by various authors, as is detailed in the methods section.

Line 119 – "model set-up … fundamentally different": Why is that so? What changes? What are the effects on final results?

- **Answer:**
    o Please see previous replies on this topic

Lines 141-142 – "… 6 cm thick layer of fine quartz sand … representing a 20 km brittle upper crust": If 60 mm in the model correspond to 20E6 mm in nature, then L* = 3E-6. This means that 1 mm in the model equals 333 m in nature. Given that the average graben in your experiments is ca. 20 mm deep, this scales up to nature to 20x333 = 6660 m, which is more than 4 times the 1600 m proposed by de Castro and Branco (1999)

- **Answer:**
  - The models cannot (and are not expected to) perfectly reproduce every single detail of the natural example. See previous replies regarding our general modelling approach.

Lines 307-308 – "… localized strain both along the intra-graben faults …": How do you explain intra-graben inversion by orthogonal shortening? This is critical to your work.

- **Answer:** All grabens without sedimentation had intra graben reactivation during inversion due to the lack of stability that sedimentation would provide like it did in the syn-rift sedimentation experiments. This is therefore not an issue (similar observations are known from other modelling works → see the Zwaan et al. 2022 review paper for more details).

Figs. 4 and 5 – several features can be measured on the topography graphs, which deserve explanation.

- **Answer:**
  - This is not the scope of our study, we are interested in the large-scale model structures, which show that inversion of initial rift faults does not really happen. Instead, reverse faulting outside of the basin are more likely to have caused uplift, an interpretation supported by field data.

Fig. 6 – panels g and h show that the graben has vanished from the rift to the inversion phases; how do you explain this? Besides, there is good evidence in panel f for inversion of the N master rift fault (sharp step in blue shades).

- **Answer:**

- The grabens did not vanish; they were filled with sedimentation.
- In the DEM, what might look like graben inversion, is an artefact of graben sedimentation. When inspecting the DIC data, it is clear there is no tracible fault reactivation.

Fig. 7 – in panels a and b you must give the references of the syn-rift layers on both sides of the faults so that we can evaluate the amount of inversion.

- **Answer:** This is not possible because there is no normal fault inversion in these models. When inspecting figures 6d,e and 8d,e, the DIC analyses only shows strain localization along the new reverse faults. Therefore, there is no inversion in the rift faults to be measured. That is sufficient observation to support our interpretation of basin inversion along newly formed reverse faults in the Araripe Basin.

Lines 377-378 – "… while no reactivation is visible in the inherited rift structures": Then how do you explain that the initial graben (panel g) has vanished (panel h) in Fig. 6? The same applies to Fig. 8.

- **Answer:** Again, the graben was filled with syn-rift sedimentation. Thus, no graben structures can be seeing in top view images during inversion.

**Abstract**

- **Author comment:** Thanks for the detailed comments on the abstract. However, we must point out that we cannot add all the details the reviewer requests in the abstract, that is what the main text is for.

**Line 15**

- **Comment:** Given like this, this 1000 m altitude means very little, because the basement could also be at 10000 m. The rpoblem is that the AB peaks at 500 m altitude above the host basement
- **Answer:** Thanks for the suggestion, it is modified.

**Line 16**

- **Comment:** 1000 m is not a topographic high if everything around is also at 1000 m

- **Answer:** We understand that, however, in this case, we say this because it is a topographic high.

**Line 18**

- **Comment:** and differential erosion

- **Answer:** Than you for the suggestion, it is modified.

**Line 18**

- **Comment:** Where are the seismic data?

- **Answer:** The seismic data observations are from works referenced in the main text (citations are not appropriate to have in the abstract)

**Line 18**

- **Comment:** Newly formed reverse faults and reactivation of precursory normal faults fully explain the field data collected by Marques et al. (2014)

- **Answer:**
    - There is very little reactivation, if at all, of precursory normal faults, as shown by seismic lines.

**Line 19**

- **Comment:** To do this, you need to: (1) reproduce the natural example in the rift phase, which you did not accomplish entirely, (2) use the full range of shortening directions relative to main boundary fault, and (3) test fault rock rheological properties

- **Answer:**

o See general reply on earlier comments on our modelling approach. We are interested in the large-scale structures, and our results fit well with field observations and seismic data.

**Line 21**

- **Comment:** Has this been observed in nature? What is the relevance of this model?
- **Answer:**
  o When running analogue models, it is important to systematically explore the parameter space so that we may understand the impact of specific parameters. As such, we added these models. (See also previous replies)

**Line 22**

- **Comment:** Should be placed between "extension" and "followed"
- **Answer:** Thanks for the suggestion, it is modified.

**Line 23**

- **Comment:** Irrelevant for an Abstract
- **Answer:** Thanks for the suggestion, we agree that this detail is not key to the abstract and can be remove. It is modified.

**Line 28**

- **Comment:** This has long been shown by previous work. "is" should be changed to "was" for verb tense consistency
- **Answer:** Thanks for the suggestion, we modified it
  o The fact that previous modelling results are reproduced with our new set-up shows the robustness of our modelling results.
  o It is not a bad thing to rerun previous models, especially since we apply new techniques (topography analysis + DIC).
  o See also previous replies

**Line 29**

- **Comment:** Normal or reverse?
- **Answer:** Reverse

**Line 29**

- **Comment:** Has this been observed in nature? What is the relevance of this model?
  **Answer:** Parameter space exploration (See previous comment). We need to understand what our models are doing, otherwise we cannot properly apply their results.

**Line 30**

- **Comment:** Images of experiments show otherwise
- **Answer:** No, they don't. Figures 6 (d,e) and 8 (d,e) do not show any rift related fault reactivation, strain is concentrated in the new reverse faults only.

**Line 32**

- **Comment:** Do you have an explanation for this behaviour?
- **Answer:** It is a mass of sand filling in the graben, buffering it from reactivation as it removed the weakness. We have add a couple of words to clarify this here.

**Line 33**

- **Comment:** This comparison with nature is misleading
- **Answer:** We do not really follow this comment. What is states in this sentence is that we propose an alternative scenario for the evolution of the Araripe Basin.

**Line 34**

- **Comment:** Where are they?
  - This refers to evidence of low-angle reverse faults outside of the basin. The evidence for this is detailed in the main text.

**Line 48**

- **Comment:** This is not true, because the main boundary fault is E-W, by reactivation of the Precambrian Patos Shear Zone. In your experiments, these are not even faults, they are "strain bands" as you call them
- **Answer:** Yes, the Patos shear zone is reactivated, however rift structures are in the Araripe Basin are mainly NE-SW. One does not invalidate the other.

**Line 49**

- **Comment:** formation?
- **Answer:** Thanks for the suggestion, it is modified.

**Line 50**

- **Comment:** remains
- **Answer:** Thanks for the suggestion, it is modified.

**Line 61**

- **Comment:** Coblentz and Richardson (1996) should also be cited here
- **Answer:** Thanks for the suggestion, it is added.

**Line 67**

- **Comment:** Only two and very short seismic lines
- **Answer:** The seismic lines showed in Rosa et al (2023) are good enough to show at least 6 rift related faults and some flower structures. The positive flower structures are interpreted as rift inversion since they do not propagate to younger

units. These seismic lines should reveal rift faults inversion affecting younger units of the Araripe basin if it had undergone major basin inversion.

**Line 67**

- **Comment:** This reference is missing in the references list, and I could not even have access to it through my Brazilian colleagues. If we cannot have access the these seismic data, you cannot use them as argument
- **Answer:** Thank you for pointing the reference is not on the list, we fixed that. Ponte & Ponte-Filho (1996) can be found online, and it is a well-known and often-cited Araripe Basin work (for instance, Marques et al (2014) cited this work as well), so we do not follow why we should not be allowed to cite it too. The fact that the data are perhaps not that easily accessible does not make them untrue after all.

**Line 77**

- **Comment:** Where is the novelty?
- **Answer:** The novelty is that we use a set-up that has previously been used for (oblique) rifting modelling for basin inversion modelling. Except for the recent efforts by Guillaume et al. (2022), this kind of set-up has never been used for basin inversion modelling (see also the Zwaan et al. 2022 review on analogue modelling of basin inversion, published in Solid Earth).
- See also previous replies.

**Line 79**

- **Comment:** I do not understand this rationale, because these angles are long known to work against tectonic inversion in high-angle normal faults
- **Answer:** We use a new set-up here, and we need to explore the parameter space. As such, we need to include orthogonal kinematics if we want to properly understand oblique kinematics in these models.

**Line 85**

- **Comment:** This is not consistent with Marques et al. (2014)
- **Answer:** We are representing the same normal faults and stratigraphic inversion sketched by Marques et al (2014) in their figure 18D.

**Line 119**

- **Comment:** Why is that so?
- **Answer:** See previous reply on the set-up

**Line 142**

- **Comment:** If 60 mm in the model correspond to 20E6 mm in nature, then $L^* = 3E-6$. This means that 1 mm in the model equals 333 m in nature.
- **Answer:** That is correct

**Line 183**

- **Comment:** In all topography graphs you must say how much you have exaggerated the vertical scale relative to the horizontal
- **Answer:** Thank you for the suggestion, we added the information to the caption.

**Line 183**

- **Comment:** Add (no sedimentation)
- **Answer:** Thank you for the suggestion, we added the information.

**Line 183**

- **Comment:** Add (with sedimentation)
- **Answer:** Thank you for the suggestion, we added the information.

**Line 307**

- **Comment:** How do you explain this inversion?
- **Answer:** Because there was no sedimentation the rift faults were not stabilized by sedimentary infill, so that the basin represented a weakness that was reactivated during compression of the model.

**Line 317**

- **Comment:** How do you expalin this?
- **Answer:** Orthogonal inversion adds more compression to the model than oblique inversion, and this translates in higher elevation for the orthogonal inversion models.

**Line 321**

- **Comment:** Add (no syn-rift sedimentation)
- **Answer:** Thank you for the suggestion, we have added it here, and also to the title of header 3.2.

**Line 360**

- **Comment:** syn-rift sedimentation
- **Answer:** Thank you for the suggestion, we made the modification.

**Line 368**

- **Comment:** at
- **Answer:** Thank you for the suggestion, we corrected that

**Line 377**

- **Comment:** Then how do you explain that the initial graben (panel g) has vanished (panel h)?
- **Answer:** The graben was filled with sediments, covering the graben structures (see previous replies on this topic).

**Line 392**

- **Comment:** This inverted fault should be drawn on the topo profile in panel h.
- **Answer:** The inverted faults are pointed to in the DIC result map view images, in the profiles and in the cross-sections shown in Figure 7.

**Line 401**

- **Comment:** What are the equivalent layers on this block? Without the references on both sides of the faults, we cannot evaluate displacements
- **Answer:** There are no equivalent layers outside basin since we only applied sediments within the basin itself. However, the thickness of each sediment layer does show the subsidence of the basin.

**Line 423**

- **Comment:** How do you explain graben vanishing?
- **Answer:** The grabens do not "vanish", they are simply being filled with syn-rift sedimentation.

**Line 457**

- **Comment:** Why do you discuss this?
- **Answer:** I discuss this because it is part of my results and an important part of my models. Leaving out a discussion of the rifting phase in the models would simply not do.

**Line 476**

- **Comment:** Where are they?

- **Answer:** The evidence is discussed in the preceding sentences. To avoid confusion, we added the references used in the discussion here as well, and slightly modified the wording.

**Line 582**

- **Comment:** Who saw these faults in the field?
- **Answer:** These are the proposed faults according to the models. We added minor normal fault inversion and concentrated deformation along new reverse faults. As we point out in the discussion, there are good grounds to predict such faults, based on field data and our model results. We now added that also Marques et al. (2014) observed some reverse faulting in the area that was not related to normal faults, which supports our interpretation of Araripe Basin inversion, even though these authors clearly champion an interpretation that is dominated by normal fault reactivation.

**Line 585**

- **Comment:** This is not consistent with data from the literature (cf. Marques et al., 2014)
- **Answer:** See previous answer, we propose a new scenario for inversion in the Araripe Basin, based on both our model results and field evidence cited in the discussion. It should not come as a surprise that our new scenario, which involves limited rift fault inversion and the establishment of new reverse faults, contradicts the scenario championed by Marques et al. (2014), which proposed major rift fault inversion as the main inversion mechanism.

**Line 485**

- **Comment:** Where did the authors get this orientation? E-W is highly oblique to the main boundary fault, the Patos shear zone, which is very different from the angles used in the experiments
- **Answer:** We here adopt the exact same orientation as Fig. 18 in Marques et al. (2014), to better illustrate the differences between both the Marques et al. 2014

and our scenario. As specified before, the Patos shear zone may delineate the northern extent of the present-day Araripe Basin domain, but the general rift basins within the Araripe Basin are oriented NE-SW, so the orientation of our schematic sections are oriented perpendicular to these NE-SW structures.

- The shortening follows from the interaction between the Andes in the west, and the Mid-Atlantic Ridge in the east, as is specified in the intro, and the orientation should indeed be ENE-WSW instead of E-W. Thanks for noticing, we have modified it in the figure and caption, as well as in the discussion.

---

## Author Response (AR3)

Dear authors,

Thank you for submitting the 2nd revision of "Analogue modelling of basin inversion: the role of oblique kinematics and implications for the Araripe Basin (Brazil)".

This is an interesting manuscript, which stimulates the scientific debate on inversion and uplift mechanisms of the Araripe basin.

- **Answer:**
    - We thank the editor for considering our manuscript and the positive evaluation.

Based on my own reading, some additional clarifications are needed.

In particular, you claim that the analogue models involving syn-rift sedimentation and orthogonal shortening with respect to the rift axes do not show basin inversion features. However, figures showing the topography insinuate uplift along the basin bounding normal faults and thus inversion of the basins. This feature has already been highlighted by R1 in the previous round of review. In your rebuttal you explain these linear features as artifacts related to placing the final layer of syn-rift sediments. This needs to be explained in the manuscript, as their meaning is not that obvious from the DIC data in the manuscript figures and the supplementary material and the reader keeps wondering about their meaning.

- **Answer:**
    - Thank you for pointing that out, we have now clearly explained this feature that might be confusing from the topography figures.

In your rebuttal you elaborate that the aim of the study was not to quantify the amount of uplift as inferred from the analogue experiments, but to gain insight on the "large-scale structures" accommodating the uplift. Also this item needs to be spelled out and justified in the in the manuscript.

- **Answer:**
    - Indeed, we do not pretend that our model captures all minute aspects of the Araripe basin, and such detailed topography comparison would therefore not really be that useful. We have added a couple of nuances to the end of the introduction, as well as at the end of section 2.1 (model set-up) and believe that this should avoid possible confusion.

Other minor items:

L73: analogue modelling by itself is not a "new approach". Please rephrase.

- **Answer:**

o Thank you for the remark, we modified it to "additional approaches" to avoid confusion.

Figures showing maximum normal strain superposed on top-view photographs are often hard to read because of the unfortunate use of a color pallet from yellow to red. In particular the yellow denoting low strain areas is almost indistinguishable from the sand color in the background. Please change the color pallet to increase readability of the figures.

- **Answer:**
  - o In fact, the background top view pictures of the experiments are in grayscale, therefore all colors displayed in the DIC figures are purely showing the maximum or minimum normal strain results. To avoid any confusion, we have added this detail to the figure captions.

Make sure that the letters related to the figure numbers in the topography plots (eg. 6f or 6n) are readable. Black on dark blue is hard to see.

- **Answer:**
  - o Thank you for the suggestion, we modified it.

L502: Panien et al. 2005 instead of 2005b.

- **Answer:**
  - o Thank you for the suggestion, we corrected it.

L515: Discussion on conditions for fault-reactivation: here you should quote the works of Rick Sibson who stared to work on this topic in the late 80's of the previous century.

- **Answer:**
  - o Thanks for the suggestion, we have now cited Sibson (1985 and 1995) with respect to the fact that normal faults tend to not reactivate when put under orthogonal compressional stresses.

Looking forward to receiving the revised version of the manuscript.

Ernst Willingshofer